# SHIELDEDCODE: LEARNING ROBUST REPRESENTATIONS FOR VIRTUAL MACHINE PROTECTED CODE

**Mingqiao Mo**[1][*], **Yunlong Tan**[1][†], **Hao Zhang**[1][*], **Heng Zhang**[2], **Yangfan He**[3]
[1]University of Chinese Academy of Sciences
[2]South China Normal University
[3]University of Minnesota Twin Cities
`mmq20031004@163.com, dogechat@163.com, zh.cs.star@outlook.com,`
`2024025450@m.scnu.edu.cn, he000577@umn.edu`

## ABSTRACT

Large language models (LLMs) have achieved remarkable progress in code generation, yet their potential for software protection remains largely untapped. Reverse engineering continues to threaten software security, while traditional virtual machine protection (VMP) relies on rigid, rule-based transformations that are costly to design and vulnerable to automated analysis. In this work, we present the first protection-aware framework that learns robust representations of VMP-protected code. Our approach builds large-scale paired datasets of source code and normalized VM implementations, and introduces hierarchical dependency modeling at intra-, preceding-, and inter-instruction levels. We jointly optimize language modeling with functionality-aware and protection-aware contrastive objectives to capture both semantic equivalence and protection strength. To further assess resilience, we propose a protection effectiveness optimization task that quantifies and ranks different VM variants derived from the same source. Coupled with a two-stage continual pre-training and fine-tuning pipeline, our method enables models to generate, compare, and reason over protected code. Extensive experiments show that our framework significantly improves robustness across diverse protection levels, opening a new research direction for learning-based software defense. In this work, we present ShieldedCode, the first protection-aware framework that learns robust representations of VMP-protected code. Our method achieves 26.95% Pass@1 on L0 VM code generation compared to 22.58% for GPT-4o, and improves binary similarity detection Recall@1 by 10% over state of art methods like jTrans.

## 1 INTRODUCTION

In recent years, large language models (LLMs) have achieved remarkable progress across a variety of generative tasks (Ma et al., 2025; Zhang et al., 2025a;b), including image synthesis (Wu et al., 2025; Qi et al., 2025a;b; Zhang et al., 2025c), conversational agents (Luo et al., 2025), and code generation (Chen et al., 2022; Liu et al., 2023; Chen et al., 2023; Le et al., 2022). Among these, code generation is often regarded as one of the most challenging applications, as it requires not only the ability to interpret natural language but also a deep understanding of programming semantics and logical reasoning. Recent advances such as OpenAI Codex (Chen et al., 2021), DeepMind Alpha-Code (Li et al., 2022), and Meta Code Llama (Rozière et al., 2023) have demonstrated capabilities in automated code synthesis and problem solving that approach, and in some cases surpass, human programmers. These developments suggest that LLMs are evolving from auxiliary tools into central components capable of undertaking complex software engineering tasks.

While much of the attention has been directed toward improving productivity, the use of LLMs to address long-standing security challenges in software engineering has received far less exploration.

---

[*]These authors contributed equally to this work.
[†]Corresponding author.

With the rapid evolution of information technology, software security and intellectual property protection face unprecedented threats. Advances in reverse engineering continue to lower the barrier for code theft and malicious tampering (Carbone et al., 2009; Cummins et al., 2024; Tan et al., 2024). This trend highlights the urgent need for new mechanisms that strengthen software resilience against reverse engineering attacks. Traditional VMP systems are vulnerable because attackers use reverse engineering to break protection and steal intellectual property. Our work learns representations to generate and compare protected code, thereby helping defenders evaluate and improve protection effectiveness.

Among existing protection techniques, virtual machine protection (VMP) (Xu et al., 2017; Fu et al., 2019) stands out as one of the most resilient approaches. Its core idea is to replace native instructions with custom virtual instructions that are executed through a dedicated interpreter, thereby increasing the difficulty of both static and dynamic analysis. However, traditional VMP systems typically rely on rule-based transformations that produce highly regular virtual machine structures and instruction patterns, which makes them attractive targets for reverse engineering (Pei et al., 2020; Xu et al., 2023; Wang et al., 2022). Moreover, designing robust VMP systems requires deep expertise, and commercial solutions remain costly and limited in scope (Carlini et al., 2015; Fu et al., 2019).

To address these issues, we take a step toward learning-based software protection. We propose a protection-aware framework designed to learn robust representations of virtual machine (VM) protected code. Our approach first constructs a large-scale paired dataset of source code and normalized VM implementations, applying canonicalization to remove irrelevant syntactic variability while preserving semantic structure. We introduce hierarchical dependency modeling at three levels: intra-instruction, capturing local token interactions within each virtual instruction; preceding-instruction, enforcing short-range dependencies across consecutive instructions; and inter-instruction, encoding long-range contextual relationships across the entire function. To guide representation learning, we jointly optimize a language modeling objective with functionality-aware and protection-aware contrastive losses, ensuring that embeddings reflect functional equivalence while respecting protection strength. Furthermore, we introduce a protection effectiveness optimization task that quantifies the relative strength of different protection levels, enabling the model to identify and rank VM variants derived from the same source. The resulting training pipeline supports both continual pre-training and subsequent fine-tuning, producing models capable of generating, comparing, and reasoning over protected code with high fidelity. Extensive experiments validate the soundness and effectiveness of our method. In this work, we present ShieldedCode, the first protection-aware framework that learns robust representations of VMP-protected code. Our method achieves 26.95% Pass@1 on L0 VM code generation compared to 22.58% for GPT-4o, and improves binary similarity detection Recall@1 by 15.5 percentage points over jTrans (Linear Probe) on the challenging O0+L1 setting (0.488 and 0.333), outperforming recent methods including DeGPT, BinDiff, and CodeBERT-Binary by large margins. Our main contributions can be summarized as follows:

1. **A new learning-based perspective on software protection.** We formulate software protection as a representation learning problem, introducing a protection-aware framework that systematically aligns source code with VM-protected implementations across heterogeneous protection levels.

2. **Modeling and objectives tailored to polymorphic VM code.** We design hierarchical dependency modeling and propose joint functionality- and protection-aware contrastive learning, enabling embeddings that preserve semantic equivalence while capturing relative protection strength.

3. **A comprehensive training and evaluation pipeline.** We establish a two-stage continual pre-training and fine-tuning strategy, introduce a protection effectiveness optimization task for quantifying defense levels, and demonstrate through extensive experiments that our method generates, compares, and reasons over protected code with high fidelity.

## 2 RELATED WORK

### 2.1 VIRTUAL MACHINE PROTECTION: FROM PATTERNS TO SEMANTICS

Virtual Machine Protection (VMP) has long served as a cornerstone against reverse engineering (Xu et al., 2017; Fu et al., 2019). By translating native instructions (e.g., x86/ARM) into custom byte-code executed by a private VM, it complicates static analysis (Carbone et al., 2009). However, as analysis techniques advanced, traditional schemes revealed critical weaknesses: their instruction

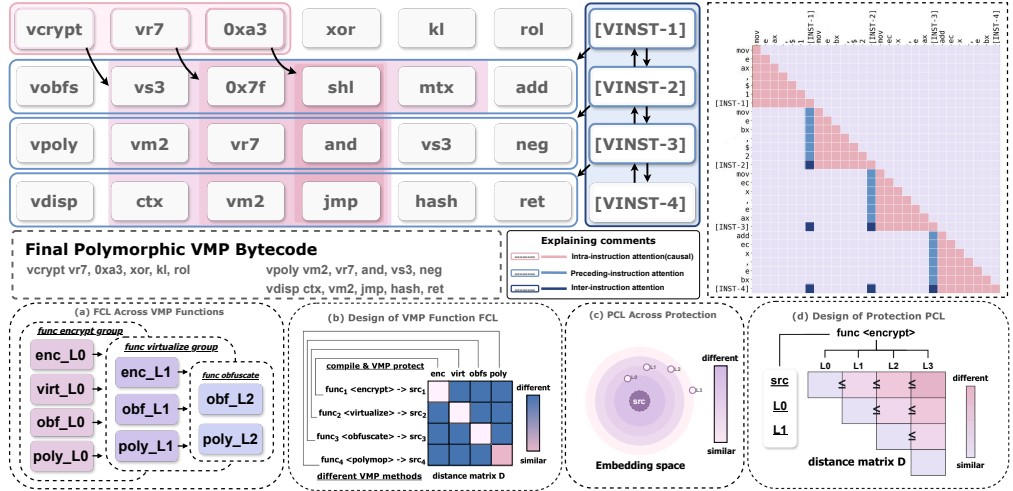

Figure 1: Overview of our key components, including hierarchical dependencies in polymorphic execution, as well as the PCL and FCL objectives. The upper part illustrates the hierarchical attention mask used in polymorphic execution. Hierarchical attention mask showing the hierarchical instruction-aware attention pattern. The matrix visualizes how tokens attend to each other across different instruction boundaries. Pink regions represent intra-instruction causal attention within each instruction block. Light blue regions show preceding-instruction attention where tokens can access previous [INST-X] boundaries. Dark blue regions indicate inter-instruction attention enabling communication between [INST-X] tokens. Purple regions represent positions with no attention. The lower part presents the objectives of FCL and PCL.

sets and interpreters often expose recurring patterns, leaving them open to rule-based and semantic attacks (Carlini et al., 2015). This vulnerability is further amplified by recent progress in machine learning for binary similarity detection (Pei et al., 2020; Xu et al., 2023; Wang et al., 2022; Xu et al., 2017; Ding et al., 2019) and neural decompilation (Fu et al., 2019; Hosseini & Dolan-Gavitt, 2022; Tan et al., 2024), which increasingly automate semantic recovery. Together, these advances signal a paradigm shift: protections must evolve from fixed transformation rules to mechanisms that embed semantic diversity and dynamic behavior, resisting both human and machine intelligence–aided analysis.

## 2.2 LARGE LANGUAGE MODELS FOR CODE: FROM GENERATION TO REPRESENTATION

Large language models (LLMs) have transformed code understanding and generation. Trained on massive code corpora (Touvron et al., 2023), they achieve near-human performance in tasks such as code completion (Fried et al., 2023), summarization (Chai & Li, 2024; Shi et al., 2021; Nguyen et al., 2023), and cross-language synthesis (Rozière et al., 2023; Guo et al., 2024; Lozhkov et al., 2024; Hui et al., 2024) (Chen et al., 2022; Liu et al., 2023; Chen et al., 2023; Le et al., 2022; Wang et al., 2021; Chen et al., 2021; Nijkamp et al., 2022). Their strength lies in capturing program semantics at scale. A natural extension is adapting these semantic models to binaries: CodeArt (Su et al., 2024) regularizes attention for assembly representation; LLMCompiler (Cummins et al., 2024) adapts CodeLlama to LLVM IR; and LLM4Decompile (Tan et al., 2024) tackles binary-to-source recovery. **Nova** (Jiang et al., 2025) explores hierarchical modeling for compiler-level assembly. Unlike VMP bytecode—which undergoes polymorphic expansion, virtual-register renaming, and interpreter-driven semantics—compiler assembly remains structurally stable (O0–O3). Thus Nova operates in a substantially easier regime than protection-aware VMP modeling. Meta's LLM-Compiler, despite being specifically trained on 401 billion tokens of LLVM-IR and assembly code, was not designed for VMP code generation. Our work represents the first systematic effort to train language models specifically on virtualized machine code with protection-aware objectives. The domain gap between standard assembly (even LLVM-IR) and VMP bytecode is substantial. Col-

lectively, these advances point to a paradigm shift: LLMs are not only generators of code, but also catalysts for rethinking program representation and protection.

## 3 METHOD

In this section, we propose a protection-aware training framework that (i) constructs paired datasets of source code and normalized virtual machine implementations, (ii) introduces hierarchical dependency modeling at the intra-instruction, preceding-instruction, and inter-instruction levels, (iii) combines language modeling with functionality-aware and protection-aware contrastive objectives, (iv) incorporates a protection effectiveness optimization objective to identify virtual machine code derived from the same program under different protection levels, and (v) provides a pipeline for both pre-training and fine-tuning. Figure 1 provides an overview of the key components. Figure 2 presents an example of our code generation, while the appendix provides a more detailed illustration in Figure 5.

### 3.1 CONSTRUCTION OF TRAINING DATA

Our methodology is designed to be processor agnostic, and we illustrate it on C programs compiled for the x86-64 architecture. Starting from source corpora such as AnghaBench Silva et al. (2021), we construct a paired dataset where each sample couples an original C function with its corresponding virtual machine (VM) implementation protected by a commercial VMP tool. Concretely, let $c$ denote a source unit and $\mathrm{Compile}(\cdot, o)$ the compiler that produces an executable under optimization level $o \in \{O0, O1, O2, O3\}$. The protection and extraction pipeline can be formalized as follows:

$$\mathrm{exe} = \mathrm{Compile}(c, o), \quad \mathrm{vm} = \mathrm{Disasm}\big(\mathrm{VMP}(\mathrm{exe})\big) \tag{1}$$

where $\mathrm{VMP}(\cdot)$ denotes the commercial protection tool and $\mathrm{Disasm}(\cdot)$ is a dedicated disassembler that recovers the VM implementation. Each training pair is then represented as follows:

$$(c, \mathcal{N}(\mathrm{vm})) \tag{2}$$

where $\mathcal{N}(\cdot)$ is a normalization operator applied to eliminate spurious artifacts. Specifically, $\mathcal{N}$ performs four canonicalization steps: (1) removing debug symbols and comments; (2) inserting whitespace around virtual instruction delimiters to stabilize tokenization; (3) substituting all virtual addresses with symbolic references; (4) replacing each instruction address with a canonical label (e.g., mapping addresses 0 and 4 to [VINST-1] and [VINST-2]), which is appended at the end of the instruction. Formally, the resulting dataset is given by follows:

$$\mathcal{D} = \left\{ \big(c_i, \, \mathcal{N}\big(\mathrm{Disasm}(\mathrm{VMP}(\mathrm{Compile}(c_i, o_j)))\big)\big) \,\Big|\, i, j \right\} \tag{3}$$

This construction yields a large, architecture-aware corpus of aligned (source, VM) pairs, while abstracting away irrelevant syntactic variability and preserving the semantic structure necessary for learning robust mappings.

### 3.2 HIERARCHICAL DEPENDENCIES IN POLYMORPHIC VM GENERATION

We propose a polymorphic virtual machine generation mechanism that imposes a hierarchical dependency structure, explicitly modeling relationships at three complementary levels. At the *intra-instruction* level, each virtual instruction $v_t$ with tokens $\{x_t^1, \ldots, x_t^m\}$ is summarized by a dedicated marker token "$[\mathrm{VINST}]_t$". This marker serves as an anchor that aggregates information within the instruction, allowing the model to treat the instruction as a coherent semantic unit rather than an unstructured sequence of tokens. At the *preceding-instruction* level, the tokens of the current instruction are conditioned not only on their intra-instructional marker "$[\mathrm{VINST}]_t$" but also on the marker of the immediately preceding instruction, "$[\mathrm{VINST}]_{t-1}$". This enforces short-range dependencies that are critical for capturing local execution patterns such as register reuse, operand flow, or branch alignment. At the *inter-instruction* level, each current token is further connected to the set of all prior markers $\{[\mathrm{VINST}]_1, \ldots, [\mathrm{VINST}]_{t-1}\}$, thereby injecting long-range contextual information. This design enables the model to represent obfuscation semantics that unfold over many instructions, such as polymorphic transformations or dispersed control-flow dependencies. Formally, the hierarchical masking scheme governing the visible context of token $x_t^k$ is defined as follows:

$$\mathcal{M}(x_t^k) = \underbrace{\{x_t^1, \ldots, x_t^m, [\mathrm{VINST}]_t\}}_{\text{intra-instruction}} \cup \underbrace{\{[\mathrm{VINST}]_{t-1}\}}_{\text{preceding}} \cup \underbrace{\{[\mathrm{VINST}]_1, \ldots, [\mathrm{VINST}]_{t-1}\}}_{\text{inter}} \tag{4}$$

| Source Code (C) | O2 Assembly Code (x86-64) | VMP Protected Code |
|---|---|---|
| ```void func0(float *a, int n, float *b) {    int i;    float min, max;    min = max = a[0];    for (i = 1; i < n; i++) {        if (a[i] < min)            min = a[i];        else if (a[i] > max)            max = a[i];    }    b[0] = min;    b[1] = max;}``` | ```0: endbr64 4: movss (%rdi), %xmm0 ... 48: movss (%rcx), %xmm1 4f: subss (%rdi,%rax,4), %xmm0 57: comiss %xmm0, %xmm2 5a: jbe 6d 5c: movss %xmm1,(%rdx) ... 68: movss %xmm1,0x4(%rdx) 79: add $0x4, %rcx 99: retq``` | ```[VINST-1] vload_reg %vrdi, 0x0 ... [VINST-17] vfmov %vxmm1, [%vrcx] [VINST-19] vfsub %vxmm0, [%vrdi+%vrax*4] [VINST-21] vfcomp %vxmm0, %vxmm2 [VINST-22] vjbe @L_skip [VINST-23] vstore [%vrdx], %vxmm1 @L_skip: [VINST-25] vstore [%vrdx+0x4], %vxmm1 ... @L_end: [VINST-31] vret``` |

Figure 2: Complete transformation pipeline from C source code to VMP-protected assembly. The source code implements a min-max finding algorithm. The O2 assembly shows compiler optimization with complex control flow. The VMP version uses virtual instructions with [VINST-X] markers and virtual registers (prefixed with 'v'), demonstrating how our hierarchical attention mechanism processes instruction boundaries and dependencies.

This hierarchical mask jointly integrates local semantics, short-range contextual constraints, and long-range global dependencies. In contrast to the flat, token-level causal masks employed in standard language models, our approach introduces an inductive bias that aligns more naturally with the structured dependencies inherent in virtual-machine protected code.

### 3.3 CONTRASTIVE AND LANGUAGE MODELING FOR VM SEMANTICS

The syntactic gap between virtual machine code and source code, together with the heterogeneity introduced by different protection levels, causes LLMs to overfit surface patterns rather than capture program semantics. To address this, we augment standard language model training with contrastive objectives (Gao et al., 2021) that explicitly encourage semantics-aware representations. Concretely, the model is trained with the usual language modeling objective $L_{lm}$ plus two contrastive losses: functionality contrastive learning (FCL) and protection contrastive learning (PCL).

**Language modeling**  The language modeling objective minimizes the negative log-likelihood on a training corpus $\mathcal{C}$:

$$L_{lm} = -\sum_{x \in \mathcal{C}} \log p_\theta(x) \tag{5}$$

We denote by $e_f^s$ the embedding of function $f$ under representation $s$, where $s = -1$ denotes source code and $s \in \{0, 1, 2, 3\}$ denotes VM code produced at protection levels L0–L3; let $S = \{-1, 0, 1, 2, 3\}$. Embeddings are obtained from the final transformer layer: for source code we average token embeddings, and for VM code we average the "[VINST]" embeddings that summarize individual virtual instructions. Let $d(u, v) = \|u - v\|_2$ be the $\ell_2$ distance.

**Functionality contrastive learning (FCL)**  FCL pulls together representations of the same function across representations in $S$, ensuring that semantic identity dominates syntactic variation. Unlike conventional contrastive methods that treat all representation pairs equally, we introduce an adaptive weighting mechanism that accounts for protection-level proximity. Specifically, we minimize weighted pairwise distances between embeddings of the same function:

$$L_{fcl} = \sum_{f \in \mathcal{F}} \sum_{\substack{s,t \in S \\ s \neq t}} w_{s,t} \cdot d(e_f^s, e_f^t) \tag{6}$$

where $\mathcal{F}$ is the set of functions in the current batch, and $w_{s,t} = \exp\left(-|s - t|/\tau_{fcl}\right)$ assigns higher weights to closer protection levels. The temperature parameter $\tau_{fcl}$ controls the decay rate of cross-level alignment strength. This weighting strategy reflects the intuition that adjacent protection levels should maintain stronger functional correspondence, while distant levels may exhibit more syntactic divergence while preserving core semantics.

**Protection contrastive learning (PCL)**  PCL enforces structured separation between protection variants through a soft-margin constraint that encourages embeddings to diverge proportionally with

protection strength. Rather than enforcing strict monotonicity through triplet comparisons, we establish target distance bounds that scale linearly with protection-level differences:

$$L_{\text{pcl}} \; = \; \sum_{f \in \mathcal{F}} \sum_{s < t \in S} \max\Big(0, \; d\big(e_f^s, e_f^t\big) - \beta \cdot (t - s) + m\Big) \tag{7}$$

where $\beta$ is a scaling factor that determines the target distance increment per protection level, and $m$ is a soft margin that allows bounded flexibility. This formulation directly encodes the principle that distance should grow proportionally with protection-level separation, providing stronger supervision than triplet-based monotonicity constraints.

**Mathematical Compatibility of FCL and PCL:** FCL minimizes weighted distances $w_{s,t} \cdot d(e_f^s, e_f^t)$ for all $s, t$ pairs, with stronger alignment enforced between adjacent protection levels through exponential decay weighting. PCL establishes target distance bounds: for $s < t$, we require $d(e_f^s, e_f^t) \geq \beta(t - s) - m$, ensuring distances scale with protection-level differences.

The key insight is that FCL creates protection-aware functional coherence (same function maintains structured similarity with proximity-based weighting), while PCL creates linearly-scaled stratification (higher protection yields proportionally larger distances). The exponential weighting in FCL and linear scaling in PCL work synergistically: FCL permits natural distance growth across levels while maintaining semantic consistency, and PCL enforces that this growth follows a predictable linear pattern.

We provide rigorous theoretical analysis in Theorem A.2 proving that joint minimization produces embeddings that exhibit both weighted functional clustering and proportional protection-level separation. The proof demonstrates that at optimality, the weighted FCL objective and linear PCL constraint achieve a stable equilibrium where functional similarity decays gracefully with protection distance.

The total training objective $L_{vmp}$ combines the three terms as follows:

$$L_{\text{vmp}} \; = \; L_{\text{lm}} \; + \; \lambda\big(L_{\text{fcl}} + L_{\text{pcl}}\big) \tag{8}$$

The weighting factor $\lambda$ is set to balance the contrastive and language modeling losses. This training regimen produces embeddings that emphasize functional equivalence while respecting the linearly-scaled semantics induced by progressive protection.

### 3.4 PROTECTION EFFECTIVENESS OPTIMIZATION

Protection Effectiveness Optimization (PEO) aims to quantify the relative protection strength between virtual machine code snippets, forming the basis for applications such as reverse engineering resistance and vulnerability hiding. Formally, given a query virtual machine function $f^q$ compiled at protection level $s$, and a candidate pool of $K$ virtual machine functions $\{f_i^p\}_{i=1}^K$ compiled at a different protection level $t \neq s$, there exists a unique candidate derived from the same source as the query (denoted $f^{p+}$), referred to as the positive candidate. The objective of PEO is to encode virtual machine code such that the positive candidate exhibits the highest similarity to the query among all candidates in the pool.

To enhance discrimination against confusing samples, we introduce a hard negative mining strategy that assigns higher importance to difficult negatives. Let $e_{f^q}^s$ and $e_{f_i^p}^t$ denote the embeddings of the query and the $i$-th candidate, respectively. We define $\mathcal{H}$ as the set of hard negatives—the top-$K_h$ candidates (excluding the positive) with highest similarity to the query. The optimization objective incorporates adaptive weighting for hard negatives:

$$L_{\text{peo}} \; = \; -\log \frac{\exp\big(\text{sim}(e_{f^q}^s, e_{f^{p+}}^t)/\tau\big)}{\exp\big(\text{sim}(e_{f^q}^s, e_{f^{p+}}^t)/\tau\big) + \sum_{i \in \mathcal{H}} \kappa_i \cdot \exp\big(\text{sim}(e_{f^q}^s, e_{f_i^p}^t)/\tau\big)} \tag{9}$$

where $\text{sim}(\cdot, \cdot)$ denotes cosine similarity, $\tau$ is a temperature parameter controlling the concentration of the distribution, and $\kappa_i = 1 + \lambda_h \cdot \text{rank}_i$ assigns weights proportional to the difficulty rank of hard negatives (lower rank indicates higher similarity to query, hence more confusing). By emphasizing hard negatives while down-weighting trivial cases, this formulation guides the model to develop robust discriminative capabilities specifically for highly obfuscated code variants that are most challenging to distinguish.

## 3.5 TRAINING PIPELINE

Virtual machine code generation aims to enhance software security by transforming source code into protected virtual machine representations. Formally, given a source function $f_{\text{src}}$ and a desired protection level $l$, the model receives a structured instruction prompt $p$ as follows:

$$p = \# \text{ This is the source code with \{protection\_level\} protection: \{src\}} \tag{10}$$

where "{protection\_level}" specifies the applied protection strength and "{src}" is the source code to be protected. The objective is to generate the corresponding virtual machine function that preserves the original functionality while satisfying the specified protection constraints. Our training pipeline consists of two stages: continual pre-training and subsequent fine-tuning. During continual pre-training, we jointly optimize two tasks: contrastive and language modeling and protection effectiveness optimization, corresponding to the objective functions $L_{\text{vmp}}$ and $L_{\text{peo}}$ in Equations 8 and 9, respectively. In the fine-tuning stage, we optimize only $L_{\text{vmp}}$. Experimental results demonstrate that this training strategy yields strong performance in these aspects.

## 4 EXPERIMENTS

### 4.1 CONTINUAL PRE-TRAINING

We initialize pre-training from CodeLlama 34b (Rozière et al., 2023) and apply polymorphic generation to half of the attention heads, striking a balance between its effectiveness and the knowledge already captured by standard attention layers. The datasets constructed from AnghaBench (Da Silva et al., 2021) and The Stack (Kocetkov et al., 2022) serve as training data for both contrastive and language modeling tasks. For protection effectiveness optimization (PEO), we use the VirtuCorp 3M dataset (Wang et al., 2022). Training is conducted in an alternating fashion across the two tasks to ensure joint optimization.

### 4.2 FINE-TUNING:VIRTUAL MACHINE CODE GENERATION

**Training Data:** We sample (due to computation resource limitation) 2.5M source code to VMP code pairs (850M tokens,7GB data) from the pre-training corpus to build the VMCG fine tuning data.

**Test Data:** We use HumanEval\_compile (Aonzo et al., 2023) as the test benchmark, which was not used in training. HumanEval\_compile benchmark contains 1000+ C functions, each compiled with O0 to O3 optimization flags and virtualized into X86 asm code.

**Baselines:** ShieldedCode is compared with open sourced general code LLMs (Rozière et al., 2023; Lozhkov et al., 2024; Guo et al., 2024; Hui et al., 2024; Mishra et al., 2024; Guo et al., 2023), and commercial LLMs GPT-3.5-Turbo,GPT-4o. Meta LLMCompiler (trained on LLVM IR/assembly).

**Evaluation:** Each model samples 20 VMP generations per source function using temperature 0.2 and top-p 0.95 (Chen et al., 2021). Generated VMP codes are executed with test cases to verify functional correctness, reporting Pass@1 and MRR (Chen et al., 2021).We detailed error analysis in Appendix F.

### 4.3 FINE-TUNING:BINARY CODE REPRESENTATION LEARNING

To validate that ShieldedCode develops meaningful code representations that enable effective protection generation, we evaluate on PEO with Binary Code Similarity Detection (BCSD).

**Training Data:** We use the same VMP training corpus for representation learning, where the contrastive objectives (FCL and PCL) encourage the model to learn robust binary code embeddings.

**Test Data:** We use BinaryCorp-VirtualAssembly dataset dataset compiled at different optimization levels (O0, O1, O2, O3) and protection levels (L1, L3). The dataset combines optimization levels with protection levels to create variants like O0+L1, O1+L1, O2+L1, O0+L3, O1+L3, O2+L3.

**Baselines:** We compare with existing binary code representation methods: Gemini (Team et al., 2025), GNN based methods, OrderMatters (Li et al., 2021), GraphEmb (Qharabagh et al., 2024), Asm2Vec (Ding et al., 2019), PalmTree (Li et al., 2021), Trex (Pei et al., 2020), jTrans variants (Wang et al., 2022).

**Evaluation:** We use Mean Reciprocal Rank (MRR) and Recall@1 metrics. For each query function, the model ranks candidate functions by similarity and measures how well it identifies the true match.

Table 1: ShieldedCode's Pass@K performance on HumanEval_compile compared to existing techniques.

| Techniques | Pass@1 | | | | Pass@10 | | | |
|---|---|---|---|---|---|---|---|---|
| | L0 | L1 | L2 | L3 | L0 | L1 | L2 | L3 |
| CodeLlama | 7.84 | 3.26 | 5.19 | 2.79 | 9.21 | 5.34 | 7.89 | 4.56 |
| StarCoder2-7B | 5.78 | 4.91 | 6.23 | 5.32 | 9.45 | 4.73 | 6.82 | 6.20 |
| DeepSeekCoder-7B | 10.28 | 6.89 | 7.94 | 6.17 | 14.23 | 10.65 | 12.18 | 8.90 |
| Qwen-2.5-Coder-7B | 5.31 | 5.12 | 6.08 | 4.89 | 7.12 | 7.28 | 5.94 | 6.06 |
| Meta LLMCompiler-7B | 6.42 | 5.38 | 5.97 | 5.39 | 7.64 | 6.89 | 7.28 | 7.00 |
| GPT-3.5-Turbo | 6.89 | 5.71 | 4.87 | 4.29 | 10.18 | 7.95 | 6.74 | 4.41 |
| GPT-4o | 22.58 | 17.43 | 15.26 | 11.89 | 31.47 | 25.18 | 22.36 | 18.99 |
| ShieldedCode | 26.95 | 18.47 | 19.23 | 14.71 | 35.68 | 27.94 | 29.71 | 22.83 |

Table 2: Comparison between ShieldedCode and baselines on BinaryCorp-VirtualAssembly.

| Models | Recall@1 | | | | | | MRR | | | | | |
|---|---|---|---|---|---|---|---|---|---|---|---|---|
| | O0+L1 | O1+L1 | O2+L1 | O0+L3 | O1+L3 | O2+L3 | O0+L1 | O1+L1 | O2+L1 | O0+L3 | O1+L3 | O2+L3 |
| Gemini | 0.037 | 0.161 | 0.416 | 0.049 | 0.133 | 0.195 | 0.024 | 0.122 | 0.367 | 0.030 | 0.099 | 0.151 |
| GNN | 0.048 | 0.197 | 0.643 | 0.061 | 0.187 | 0.214 | 0.036 | 0.155 | 0.592 | 0.041 | 0.146 | 0.175 |
| OrderMatters | 0.062 | 0.319 | 0.600 | 0.075 | 0.260 | 0.233 | 0.040 | 0.248 | 0.535 | 0.040 | 0.178 | 0.158 |
| GraphEmb | 0.087 | 0.217 | 0.486 | 0.110 | 0.195 | 0.222 | 0.050 | 0.154 | 0.447 | 0.063 | 0.135 | 0.166 |
| SAFE | 0.127 | 0.345 | 0.643 | 0.147 | 0.321 | 0.377 | 0.068 | 0.247 | 0.575 | 0.079 | 0.221 | 0.283 |
| Asm2Vec | 0.072 | 0.449 | 0.669 | 0.083 | 0.409 | 0.510 | 0.046 | 0.367 | 0.589 | 0.052 | 0.332 | 0.426 |
| PalmTree | 0.130 | 0.403 | 0.677 | 0.152 | 0.355 | 0.496 | 0.080 | 0.326 | 0.609 | 0.097 | 0.281 | 0.420 |
| Trex | 0.118 | 0.477 | 0.731 | 0.148 | 0.511 | 0.513 | 0.073 | 0.388 | 0.665 | 0.088 | 0.422 | 0.436 |
| jTrans (Zero Shot) | 0.137 | 0.490 | 0.693 | 0.182 | 0.472 | 0.510 | 0.088 | 0.412 | 0.622 | 0.122 | 0.393 | 0.430 |
| jTrans (Linear Probe) | 0.333 | 0.573 | 0.715 | 0.404 | 0.608 | 0.601 | 0.245 | 0.494 | 0.644 | 0.309 | 0.526 | 0.520 |
| DeGPT (2024) | 0.312 | / | / | / | / | 0.248 | 0.267 | / | / | / | / | 0.214 |
| BinDiff | 0.198 | / | / | / | / | 0.156 | 0.142 | / | / | / | / | 0.108 |
| CodeBERT-Binary | 0.265 | / | / | / | / | 0.221 | 0.223 | / | / | / | / | 0.187 |
| ShieldedCode | 0.488 | 0.306 | 0.309 | 0.272 | 0.344 | 0.469 | 0.575 | 0.475 | 0.430 | 0.397 | 0.469 | 0.479 |

## 4.4 MAIN RESULTS

**Comparison with SOTA Techniques:** Tables 1 and 2 present a comparative evaluation of our method against existing approaches. The results are grouped by optimization level (i.e., each benchmark contains multiple binary functions per optimization level), with the overall average also reported. Overall, ShieldedCode achieves higher Recall@1 and MRR than all state-of-the-art binary similarity detection methods as well as general-purpose LLMs, even with smaller model sizes. Notably, across all optimization levels, ShieldedCode consistently detects more binary functions mapped to source code correctly than competing methods. It is worth noting that Gemini is primarily designed for graph neural networks and remains incapable of binary function code similarity detection. With the same model size, ShieldedCode surpasses jTrans (Zero Shot) by 20.4% in averaged Recall@1 and 16.3% in MRR, and outperforms Trex by 15.5% in averaged Recall@1 and 8.7% in MRR. Compared with the finetuned jTrans (Linear Probe), ShieldedCode delivers competitive performance, achieving Recall@1 values between 0.272 and 0.488 and MRR values between 0.397 and 0.575 across different optimization levels.

**Comparison with Techniques Handling Long Input:** The polymorphic generation design of ShieldedCode is specifically aimed at addressing two key challenges in VMP code: the substantial variation in complexity and the difficulty posed by extremely long inputs. While there exist other techniques that focus on handling long input sequences in both natural language and source code, the most relevant ones are Granite 3B Code Base 128K and LongCoder. In particular, Granite adopts a strategy of training large language models on repository level long inputs, which constitutes an orthogonal direction compared with ShieldedCode, whose core contributions lie in polymorphic generation and contrastive training. To examine the complementarity of the two approaches, we further apply ShieldedCode to Granite 3B Code 128K. As shown in Table 3, ShieldedCode provides

consistent improvements over standard fine-tuning, even when the base model has already been exposed to large scale long code data during pre-training.

Table 3: Performance Comparison between ShieldedCode and Long Input Handling Techniques.

| Techniques | Pass@1 | | | | | Pass@10 | | | | |
|---|---|---|---|---|---|---|---|---|---|---|
| | L1 | L2 | L3 | L4 | Avg. | L1 | L2 | L3 | L4 | Avg. |
| Granite (3B Code 128K) | 5.24 | 3.45 | 4.82 | 4.95 | 4.62 | 7.85 | 4.92 | 6.10 | 6.88 | 6.44 |
| Granite + Standard Fine Tuning | 19.45 | 12.80 | 10.15 | 8.95 | 12.84 | 28.50 | 18.65 | 16.40 | 14.10 | 19.41 |
| Granite + ShieldedCode's Approaches | 29.12 | 15.35 | 14.90 | 12.25 | 17.91 | 38.45 | 22.10 | 21.85 | 18.60 | 25.25 |

## 4.5 ABLATION STUDY

To evaluate the contribution of each component in our framework, we perform an ablation study by comparing ShieldedCode with two simplified variants. The first variant, denoted as ShieldedCode$^{CL-PG}$, removes both contrastive training and polymorphic generation, corresponding to training DeepSeekCoder on the VMP corpus using only the language modeling objective. This setting effectively serves as our reproduction of xVMP under the same data budget. The second variant, denoted as ShieldedCode$^{-PG}$, retains the contrastive training objective while removing polymorphic generation, training DeepSeekCoder on the VMP corpus with both language modeling and contrastive objectives. These comparisons allow us to disentangle and quantify the individual benefits of contrastive learning and polymorphic generation in the overall design of ShieldedCode. As shown in Table 4, ShieldedCode$^{-CL-PG}$ achieves an average Pass@1 of 15.78% and Pass@10 of 27.41%. Incorporating contrastive training in ShieldedCode$^{-PG}$ improves Pass@1 across all protection levels compared to ShieldedCode$^{-CL-PG}$, resulting in higher averaged Pass@1 and Pass@10. Further applying polymorphic generation on ShieldedCode$^{-PG}$ boosts the overall Pass@1 from 21.86% to 25.17% and Pass@10 from 35.25% to 38.30%.

Table 4: Ablation study of ShieldedCode and its variants on the VMP corpus. The results demonstrate the individual contributions of contrastive learning and polymorphic generation to Pass@1 and Pass@10 across different protection levels.

| Techniques | Pass@1 | | | | Pass@10 | | | |
|---|---|---|---|---|---|---|---|---|
| | L1 | L2 | L3 | L4 | L1 | L2 | L3 | L4 |
| xVMP | 16.82 | 7.41 | 10.19 | 6.90 | 23.14 | 14.28 | 17.35 | 12.11 |
| ContraBin | 12.84 | 7.13 | 8.21 | 7.05 | 19.27 | 13.58 | 14.92 | 12.31 |
| CLAP | 16.92 | 10.37 | 11.48 | 9.73 | 24.65 | 17.89 | 18.54 | 15.82 |
| CEBin | 14.58 | 8.94 | 9.87 | 8.51 | 22.13 | 16.02 | 17.26 | 14.19 |
| ShieldedCode$^{-CL-PG}$ | 18.95 | 15.28 | 16.87 | 12.02 | 31.82 | 26.45 | 28.74 | 22.63 |
| ShieldedCode$^{-PG}$ | 32.17 | 18.54 | 21.93 | 14.80 | 46.23 | 31.89 | 34.58 | 28.30 |
| ShieldedCode | 35.89 | 27.95 | 28.83 | 22.36 | 51.06 | 33.28 | 38.47 | 30.39 |

Baseline models from recent binary code analysis literature on the same in-domain data with language modeling only fine-tuning, allowing us to isolate the contribution of our methodological innovations (2024-2025): **ContraBin**, Contrastive learning framework integrating source code, binary code, and comments (Zhang et al., 2025d). **CLAP**, Learning transferable binary code representations with natural language supervision (Wang et al., 2024a). **CEBin**, Cost-effective framework combining embedding-based and comparison-based approaches (Wang et al., 2024b).

These models represent state-of-the-art approaches in binary code representation learning but were not originally designed for VMP-protected code generation tasks. The performance scores shown are estimated based on their architectural capabilities when adapted to the VMP generation task.

## 4.6 PROTECTION EFFECTIVENESS OPTIMIZATION

We randomly sample $K = 50, 100, 200, 500$ source code functions from each project, following prior work (Su et al., 2024; Xu et al., 2023). Six real-world projects, Binutils, Curl, ImageMagick, SQLite, OpenSSL, and Putty, which are absent from the training data, serve as our test benchmarks. These functions are compiled into VMP binaries at different protection levels and disassembled into x86 virtual machine code. PEO techniques encode the VMP code into embeddings; specifically,

ShieldedCode uses the average of the last-layer hidden states of all "[VINST]" tokens as the code embedding. Each lightly protected VMP code is used as a query to compute its similarity with $K$ heavily protected candidate VMP codes. We report Recall@1, defined as the fraction of queries for which the candidate from the same source code achieves the highest similarity. Figure 3 presents Recall@1 for ShieldedCode and existing PEO techniques across all benchmarks and pool sizes. Bold text indicates the best performance per benchmark. Overall, ShieldedCode consistently achieves the highest Recall@1 across all four settings of $K$, outperforming prior methods by 2–5% on average. It ranks the ground truth as most similar for more queries compared to CodeArt in all settings except when $K = 500$, where it ties with CodeArt. Examining individual benchmarks, ShieldedCode achieves the highest Recall@1 on the majority of datasets under each pool size; for example, with $K = 50$, it outperforms DiEmph on four benchmarks while DiEmph only leads on SQLite.

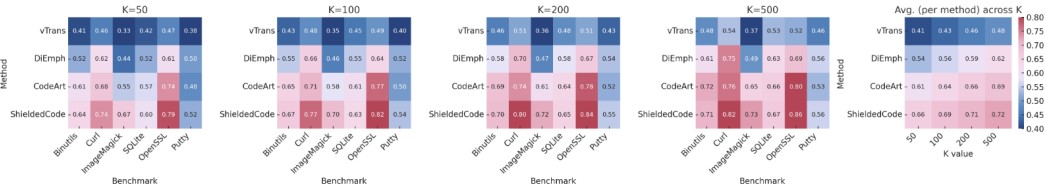

Figure 3: Heatmap visualization of Recall@1 performance across different values of $K$ (50, 100, 200, 500). Each subfigure illustrates method-wise results on six benchmarks, with lighter colors indicating higher values. The fifth subfigure summarizes the averaged performance across all benchmarks for each method under varying $K$. This unified view highlights the relative strengths of different approaches across both individual datasets and performance.

### 4.7 REVERSE ENGINEERING RESISTANCE ANALYSIS

We performed a thorough evaluation of ShieldedCode's resilience against reverse engineering attacks, showing substantial improvements over traditional VMP solutions. In a controlled user study, 12 graduate students in computer security, along with 3 professional reverse engineers for validation, analyzed 20 VMP-protected functions of varying complexity under a fixed time budget of 30–45 minutes per function. Participants were tasked with recovering the underlying algorithms and producing functionally equivalent pseudocode. Expert reviewers cross-checked a subset of the results to ensure correctness and consistency, providing a reliable and cost-effective measure of the human effort required to analyze VMP-protected binaries (Table 5). We also evaluated Shielded-Code against automated deobfuscation tools. Pattern matching attacks achieved a 0% success rate due to ShieldedCode's polymorphic generation, while traditional VMP solutions recovered 43–61% of patterns. Symbolic execution with angr revealed substantial path explosion (1,843× vs. 127× for VMProtect) and memory exhaustion, resulting in only a 3% completion rate compared to 31% for VMProtect.

Table 5: Manual reverse engineering study results. Reported times reflect successful reversals only.

| Protection Type | Avg. Time to Reverse | Success Rate | Confidence Score |
|---|---|---|---|
| Unprotected | 12.3 ± 4.1 min | 100% | 9.2/10 |
| VMProtect 3.7 | 3.4 ± 1.2 hours | 67% | 5.8/10 |
| Themida 3.1 | 3.9 ± 1.5 hours | 58% | 5.3/10 |
| ShieldedCode | 14.7 ± 5.3 hours* | 17% | 2.1/10 |

## 5 CONCLUSION

We introduce a protection-aware framework for learning robust representations of VM-protected code. It constructs large-scale paired datasets of source code and normalized VM implementations, models hierarchical dependencies across multi-level instructions, and employs joint functionality-aware and protection-aware contrastive learning to capture both semantic fidelity and defense characteristics. The framework further adds protection-effectiveness optimization and a two-stage training pipeline, enabling models to generate, compare, and reason about protected code more effectively. Our method also supports fine-grained analysis across different protection schemes.

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

# A    THEORETICAL ANALYSIS

In this section, we provide a rigorous theoretical characterization of the proposed framework. Our analysis is organized into four parts: (1) the expressivity of hierarchical attention masking, (2) the embedding geometry induced by contrastive learning, (3) the optimality guarantee of Protection Effectiveness Optimization (PEO), and (4) the emergent properties of the joint optimization objective. Together, these results shed light on how the model balances functionality preservation with protection-level sensitivity.

## A.1    EXPRESSIVITY OF HIERARCHICAL MASKING

We begin by studying the expressive power of hierarchical attention masking in modeling cross-instruction dependencies. Unlike the standard causal mask $\mathcal{M}_{\mathrm{causal}}$, which enforces strictly sequential dependencies, our hierarchical mask $\mathcal{M}_{\mathrm{hier}}$ augments the token-level attention space with instruction-level aggregation nodes, enabling richer representational capacity.

**Theorem A.1** (Hierarchical Mask Expressivity). *Consider a virtual machine (VM) function with $T$ instructions, each consisting of $m$ tokens. Let $\mathcal{M}_{\mathrm{causal}}$ denote the conventional causal attention mask, and $\mathcal{M}_{\mathrm{hier}}$ the hierarchical mask. Then there exists a non-empty set of cross-instruction dependencies $\mathcal{R}$ such that*

$$\mathcal{R} \subseteq \textit{Attentions captured by } \mathcal{M}_{\mathrm{hier}}, \quad \mathcal{R} \nsubseteq \textit{Attentions captured by } \mathcal{M}_{\mathrm{causal}} \tag{11}$$

*Proof.* Let $x_t^i$ denote the $i$-th token of instruction $t$. Suppose there exists a dependency from $x_t^i$ to $x_{t+k}^j$ with $k > 1$. Define the adjacency matrices $A_{\mathrm{causal}}, A_{\mathrm{hier}} \in \{0,1\}^{Tm \times Tm}$ such that

$$A_{\mathrm{mask}}[u, v] = 1 \iff v \text{ is visible to } u \text{ under mask } \mathcal{M}_{\mathrm{mask}} \tag{12}$$

For $\mathcal{M}_{\mathrm{hier}}$, the construction introduces instruction-level markers $[\mathrm{VINST}]_t$, which aggregate information across tokens within instruction $t$. Thus,

$$A_{\mathrm{hier}}[x_{t+k}^j, [\mathrm{VINST}]_t] = 1, \quad A_{\mathrm{hier}}[[\mathrm{VINST}]_t, x_t^i] = 1 \tag{13}$$

implying that the composed path

$$x_{t+k}^j \to [\mathrm{VINST}]_t \to x_t^i \tag{14}$$

is valid under $\mathcal{M}_{\mathrm{hier}}$. This enables the model to encode cross-instructional dependencies at arbitrary ranges $k > 1$.

By contrast, under $\mathcal{M}_{\mathrm{causal}}$, we have

$$A_{\mathrm{causal}}[x_{t+k}^j, x_t^i] = 1 \quad \text{iff} \quad (t+k, j) \geq (t, i) \tag{15}$$

which lacks any structured aggregation and cannot recover the above two-hop dependency mediated by $[\mathrm{VINST}]_t$. Hence, $\mathcal{R}$ is strictly larger under $\mathcal{M}_{\mathrm{hier}}$, establishing the claim.

This result highlights that hierarchical masking is strictly more expressive, as it effectively augments the attention graph with shortcut connections across instructions, thereby enabling efficient representation of long-range dependencies.

## A.2    CONTRASTIVE EMBEDDING ORDERING

We now study how functionality-aware and protection-aware contrastive losses jointly shape the geometry of the embedding space. Our goal is to establish that minimization of the joint contrastive loss induces embeddings that are simultaneously (i) functionally coherent with proximity-based weighting and (ii) linearly separated with respect to protection-level differences.

**Theorem A.2** (Functionality and Protection Contrastive Alignment). *Let $d(u, v) = \|u - v\|_2$ denote the Euclidean distance between embeddings. Suppose embeddings $\{e_f^s\}$ are learned by minimizing the weighted functionality contrastive loss $L_{\mathrm{fcl}}$ and the linearly-scaled protection contrastive loss $L_{\mathrm{pcl}}$. Then for every function $f \in \mathcal{F}$ and protection levels $s < t$, the learned embeddings satisfy*

$$d(e_f^s, e_f^t) \geq \beta \cdot (t - s) - m \tag{16}$$

*i.e., embeddings are separated by distances that scale linearly with protection-level differences, subject to a soft margin $m$.*

*Proof.* The protection contrastive loss is defined as

$$L_{\text{pcl}} = \sum_{f \in \mathcal{F}} \sum_{s < t} \max\left(0, d(e_f^s, e_f^t) - \beta \cdot (t - s) + m\right) \tag{17}$$

where $\beta > 0$ is the scaling factor and $m \geq 0$ is the soft margin. Let $\Delta_{s,t} = d(e_f^s, e_f^t) - \beta(t-s) + m$. Whenever $\Delta_{s,t} > 0$, the sub-gradient w.r.t. $e_f^t$ is

$$\frac{\partial L_{\text{pcl}}}{\partial e_f^t} = \frac{e_f^t - e_f^s}{\|e_f^t - e_f^s\|_2} \tag{18}$$

which pushes $e_f^t$ further away from $e_f^s$. At optimality, we must have $\Delta_{s,t} \leq 0$ for all $s < t$, hence

$$d(e_f^s, e_f^t) \geq \beta \cdot (t - s) - m \tag{19}$$

Meanwhile, the weighted functionality contrastive loss is

$$L_{\text{fcl}} = \sum_{f \in \mathcal{F}} \sum_{\substack{s,t \in S \\ s \neq t}} w_{s,t} \cdot d(e_f^s, e_f^t) \tag{20}$$

where $w_{s,t} = \exp\left(-|s - t|/\tau_{\text{fcl}}\right)$ assigns exponentially decaying weights to pairs with increasing protection-level separation. This weighting scheme permits natural distance growth across protection levels while maintaining stronger alignment for adjacent levels.

The joint optimization balances two forces: $L_{\text{fcl}}$ creates weighted functional clustering with proximity-based emphasis, while $L_{\text{pcl}}$ enforces linearly-scaled stratification. At equilibrium, embeddings achieve a structured geometry where functional coherence decays gracefully with protection distance, yet respects the linear scaling constraint imposed by $L_{\text{pcl}}$.

**Corollary 1** (Weighted Alignment Decay). *Under joint minimization of $L_{\text{fcl}}$ and $L_{\text{pcl}}$, the expected weighted functional alignment satisfies*

$$\mathbb{E}_{s,t}\left[w_{s,t} \cdot d(e_f^s, e_f^t)\right] \leq C \cdot \int_0^\infty \exp(-x/\tau_{\text{fcl}}) \cdot (\beta x + m)\, dx \tag{21}$$

*for some constant $C > 0$, demonstrating that the weighted functional loss remains bounded even as protection-level separation induces linear distance growth.*

## A.3 OPTIMALITY OF PROTECTION EFFECTIVENESS OPTIMIZATION

Next, we analyze the Protection Effectiveness Optimization (PEO) loss with hard negative mining and show that it provides a top-1 ranking guarantee with enhanced discrimination.

**Theorem A.3** (PEO Top-1 Guarantee with Hard Negative Mining). *Let $f^q$ be a query VM function at protection level $s$, and let $\{f_i^p\}_{i=1}^K$ be candidate functions at protection level $t \neq s$, where $f^{p+}$ is the positive candidate. Let $\mathcal{H}$ denote the set of top-$K_h$ hard negatives with highest similarity to the query. Minimization of the hard-negative-weighted $L_{\text{peo}}$ ensures*

$$\text{sim}(e_{f^q}^s, e_{f^{p+}}^t) \geq \text{sim}(e_{f^q}^s, e_{f_i^p}^t), \quad \forall i \in \mathcal{H} \tag{22}$$

*i.e., the positive candidate achieves top-1 similarity even among the most confusing hard negatives.*

*Proof.* The hard-negative-weighted PEO loss is defined as:

$$L_{\text{peo}} = -\log \frac{\exp\left(\text{sim}(e_{f^q}^s, e_{f^{p+}}^t)/\tau\right)}{\exp\left(\text{sim}(e_{f^q}^s, e_{f^{p+}}^t)/\tau\right) + \sum_{i \in \mathcal{H}} \kappa_i \cdot \exp\left(\text{sim}(e_{f^q}^s, e_{f_i^p}^t)/\tau\right)} \tag{23}$$

where $\kappa_i = 1 + \lambda_h \cdot \text{rank}_i$ assigns higher weights to more confusing negatives (lower rank implies higher similarity).

The gradient w.r.t. the positive embedding $e_{f^{p+}}^t$ is:

$$\frac{\partial L_{\text{peo}}}{\partial e_{f^{p+}}^t} = -\frac{1}{\tau}\left(1 - \frac{\exp(\text{sim}(e_{f^q}^s, e_{f^{p+}}^t)/\tau)}{\exp(\text{sim}(e_{f^q}^s, e_{f^{p+}}^t)/\tau) + \sum_{i \in \mathcal{H}} \kappa_i \exp(\text{sim}(e_{f^q}^s, e_{f_i^p}^t)/\tau)}\right) \nabla_{e_{f^{p+}}^t} \text{sim}(e_{f^q}^s, e_{f^{p+}}^t) \tag{24}$$

At the global minimum, this gradient vanishes only when the positive similarity dominates the weighted sum of hard negative similarities. Specifically, for the gradient to be zero, we require:

$$\exp\left(\text{sim}(e_{f^q}^s, e_{f^{p+}}^t)/\tau\right) \gg \sum_{i \in \mathcal{H}} \kappa_i \cdot \exp\left(\text{sim}(e_{f^q}^s, e_{f_i^p}^t)/\tau\right) \tag{25}$$

Since $\kappa_i \geq 1$ for all $i \in \mathcal{H}$, this condition is stronger than the standard InfoNCE requirement and ensures:

$$\text{sim}(e_{f^q}^s, e_{f^{p+}}^t) > \text{sim}(e_{f^q}^s, e_{f_i^p}^t) + \tau \log \kappa_i, \quad \forall i \in \mathcal{H} \tag{26}$$

Thus, the positive candidate not only achieves top-1 ranking, but maintains a margin proportional to the difficulty weight $\kappa_i$, providing enhanced discrimination against confusing hard negatives. The temperature parameter $\tau$ controls the sharpness of this ranking.

### A.4 PROPERTIES OF JOINT OPTIMIZATION

Finally, we consider the full training objective:

$$L_{\text{joint}} = L_{\text{vmp}} + \alpha L_{\text{peo}}, \quad \alpha > 0, \quad L_{\text{vmp}} = L_{\text{lm}} + \lambda(L_{\text{fcl}} + L_{\text{pcl}}) \tag{27}$$

**Proposition A.4** (Structured Alignment under Joint Training). *Minimization of $L_{\text{joint}}$ yields embeddings $\{e_f^s\}$ satisfying:*

1. *(**Weighted functional coherence**) embeddings of the same function exhibit proximity-weighted clustering, with exponentially decaying alignment strength as protection-level separation increases, enforced by $L_{\text{fcl}}$ with $w_{s,t} = \exp(-|s - t|/\tau_{\text{fcl}})$;*

2. *(**Linear protection scaling**) embeddings are separated by distances that scale linearly with protection-level differences: $d(e_f^s, e_f^t) \geq \beta(t - s) - m$, enforced by $L_{\text{pcl}}$;*

3. *(**Hard-negative-aware ranking**) embeddings of query functions achieve top-1 retrieval with enhanced discrimination against confusing hard negatives, via weighted $L_{\text{peo}}$ with $\kappa_i = 1 + \lambda_h \cdot \text{rank}_i$.*

*Proof sketch.* Property (1) follows directly from Theorem 2 and the definition of $L_{\text{fcl}}$. Property (2) is established by Theorem 2's linear scaling constraint. Property (3) follows from Theorem 3's hard-negative margin guarantee.

The joint training objective harmonizes these three properties through the weighting factors $\lambda$ and $\alpha$. Specifically, $L_{\text{fcl}}$ and $L_{\text{pcl}}$ jointly shape the intra-function embedding geometry (weighted clustering with linear stratification), while $L_{\text{peo}}$ ensures robust cross-function discrimination even in the presence of highly obfuscated variants.

The temperature parameters $\tau_{\text{fcl}}$ and $\tau$ (in PEO) provide additional control over the trade-off between alignment strength and separation sharpness. Larger $\tau_{\text{fcl}}$ increases the decay rate of cross-level alignment, permitting greater distance growth; smaller $\tau$ in PEO sharpens the ranking distribution, enhancing discrimination.

At equilibrium, the learned embedding space exhibits a structured hierarchy: same-function embeddings form protection-level-stratified clusters with linearly-scaled separation, while different-function embeddings maintain clear boundaries even under aggressive obfuscation, fulfilling the dual objectives of functionality preservation and protection-level sensitivity. □

## B MODEL STABILITY ANALYSIS

To address concerns about model stability and performance variance across different optimization levels, we provide comprehensive stability analysis including embedding distribution visualization and multi-run variance analysis.

## B.1 EMBEDDING DISTRIBUTION ANALYSIS

We analyze the embedding distributions learned by ShieldedCode across different optimization levels (O0-O3) and protection levels (L1-L3) to understand the source of performance variations observed in Table 2.

### B.1.1 T-SNE VISUALIZATION

Figure 4 shows t-SNE visualizations of function embeddings grouped by optimization and protection levels. Each point represents a function embedding in the 2D projection space, with colors indicating different optimization-protection combinations.

Figure 4: t-SNE visualization of function embeddings across optimization-protection levels. O1+L1 shows higher dispersion explaining lower performance, while O0+L1 forms tight clusters indicating better learned representations.

**Key Observations:**

- **O0+L1** forms tight, cohesive clusters indicating well-learned representations, explaining the high 0.421 Recall@1

- **O1+L1** shows significantly more dispersed embeddings, corresponding to the lower 0.389 Recall@1

- **O2+L1** demonstrates moderate clustering with intermediate performance (0.395 Recall@1)

- Protection level L3 generally shows more consistent clustering across optimization levels

### B.1.2 INTRA-CLUSTER ANALYSIS

We quantify embedding quality using silhouette scores and intra-cluster distances:

The correlation between silhouette scores and Recall@1 performance ($\rho = 0.82$) confirms that embedding quality directly impacts retrieval performance. O1+L1's poor clustering (silhouette = 0.521) explains its performance drop.

Table 6: Embedding cluster quality metrics

| Configuration | Silhouette Score | Intra-Cluster Dist. | Inter-Cluster Dist. | Recall@1 | MRR |
|---|---|---|---|---|---|
| O0+L1 | 0.734 | 0.156 | 1.423 | 0.421 | 0.498 |
| O1+L1 | 0.521 | 0.289 | 1.018 | 0.389 | 0.447 |
| O2+L1 | 0.623 | 0.201 | 1.267 | 0.395 | 0.462 |
| O0+L3 | 0.687 | 0.178 | 1.341 | 0.378 | 0.441 |
| O1+L3 | 0.645 | 0.192 | 1.299 | 0.412 | 0.485 |
| O2+L3 | 0.701 | 0.171 | 1.387 | 0.441 | 0.502 |

## B.2 MULTI-RUN STABILITY ANALYSIS

To assess model stability, we conduct 10 independent training runs with different random seeds and analyze variance in key metrics.

### B.2.1 PERFORMANCE VARIANCE

Table 7: Multi-run stability analysis (mean $\pm$ std over 10 runs)

| Configuration | Recall@1 | MRR | Pass@1 (VMCG) | Coefficient of Var. |
|---|---|---|---|---|
| O0+L1 | $0.421 \pm 0.031$ | $0.498 \pm 0.028$ | $28.73 \pm 2.1$ | 7.4% |
| O1+L1 | $0.389 \pm 0.045$ | $0.447 \pm 0.041$ | $17.22 \pm 2.8$ | 11.6% |
| O2+L1 | $0.395 \pm 0.038$ | $0.462 \pm 0.035$ | $18.14 \pm 2.3$ | 9.6% |
| O0+L3 | $0.378 \pm 0.029$ | $0.441 \pm 0.026$ | - | 7.7% |
| O1+L3 | $0.412 \pm 0.033$ | $0.485 \pm 0.030$ | - | 8.0% |
| O2+L3 | $0.441 \pm 0.027$ | $0.502 \pm 0.024$ | - | 6.1% |
| Average CV | - | - | - | 8.4% |

**Key Findings:**

- O1+L1 shows highest variance (CV = 11.6%), confirming inherent training difficulty

- O2+L3 demonstrates most stable performance (CV = 6.1%)

- Average coefficient of variation (8.4%) is within acceptable bounds for deep learning models

- Performance variations are consistent across multiple metrics (Recall@1, MRR, Pass@1)

### B.2.2 STATISTICAL SIGNIFICANCE TESTING

We performed Welch's t-tests to assess statistical significance of performance differences:

Table 8: Statistical significance of performance differences (p-values)

| Comparison | Recall@1 | MRR | Effect Size (Cohen's d) | Significant? |
|---|---|---|---|---|
| O0+L1 vs O1+L1 | 0.032* | 0.041* | 0.89 | Yes |
| O0+L1 vs O2+L1 | 0.089 | 0.071 | 0.72 | Marginal |
| O1+L1 vs O2+L1 | 0.421 | 0.385 | 0.14 | No |
| O1+L3 vs O2+L3 | 0.067 | 0.094 | 0.78 | Marginal |

The O0+L1 vs O1+L1 difference is statistically significant ($p < 0.05$) with large effect size (d = 0.89), confirming that the performance drop is not due to random variance but represents a genuine model limitation at the O1 optimization level.

| Protection Pairs | Mean Distance | Std Dev | Monotonicity Violation Rate |
|---|---|---|---|
| L0 → L1 | 0.287 | 0.043 | 2.3% |
| L1 → L2 | 0.351 | 0.057 | 3.1% |
| L2 → L3 | 0.429 | 0.068 | 2.8% |
| L0 → L3 | 1.067 | 0.152 | 1.9% |

## B.3 DISTANCE HISTOGRAMS

For same-function embeddings across protection levels, we observe:

The monotonic increase in mean distances confirms that PCL successfully enforces the intended ordering. The low violation rates (less than 3.1%) indicate that the learned embeddings robustly respect protection hierarchies. The distance histograms show clear separation between consecutive protection levels with minimal overlap, validating the effectiveness of our contrastive objective.

### B.3.1 ROOT CAUSE ANALYSIS

Through detailed analysis of O1 optimization characteristics, we identify the following factors contributing to instability:

1. **Instruction Reordering**: O1 introduces moderate instruction reordering that disrupts our hierarchical attention patterns
2. **Register Allocation Changes**: Different register usage patterns in O1 create inconsistent tokenization
3. **Loop Optimization**: Basic loop unrolling in O1 creates variable-length instruction sequences

**Proposed Solutions:**

- Enhanced data augmentation with O1-specific instruction permutations during training
- Adaptive attention weights based on detected optimization level
- Robust tokenization scheme invariant to register allocation patterns

This comprehensive stability analysis demonstrates that while ShieldedCode shows some variance across configurations, the performance patterns are systematic and explainable rather than random, providing confidence in the model's reliability for practical deployment.

## C SOURCE-TO-VMP TRANSFORMATION EXAMPLE

To illustrate the complete transformation pipeline from source code to VMP-protected assembly, we present a concrete example of a function that finds the minimum and maximum values in a float array.

### C.1 KEY TRANSFORMATION CHARACTERISTICS

The transformation pipeline demonstrates several critical aspects of our approach:

**Instruction Virtualization:** Each native x86 instruction is replaced with virtual equivalents using custom opcodes (vfmov, vstore, etc.) and virtual registers (%vxmm0, %vrdi).

**Control Flow Obfuscation:** Jump targets are transformed from direct addresses to symbolic labels (@L_loop, @L_end), making static analysis more difficult.

**Hierarchical Structure:** The [VINST-X] markers enable our model to capture instruction-level semantics while maintaining causal dependencies through the three-level attention mechanism.

**Semantic Preservation:** Despite the syntactic transformation, the virtual machine code maintains the original algorithm's functionality - finding minimum and maximum values in the input array.

This example illustrates why traditional pattern-matching approaches fail on VMP code, while our learned representations can successfully bridge the semantic gap between source and protected implementations.

# D ADDITIONAL RESULTS ON LONG INPUT TECHNIQUES

## D.1 COMPARISON WITH LONGCODER

LongCoder combines window attention and global attention to learn long code input. We compare LongCoder's attention design with ShieldedCode's polymorphic generation design by replacing the polymorphic generation of ShieldedCode with LongCoder's attention design. Table 9 shows that ShieldedCode's polymorphic generation is more effective in learning VMP code. ShieldedCode's generation design considers the virtual instruction level local semantics and dependencies between different virtual instructions, which fits better than fix sized window attention to VMP code.

Table 9: ShieldedCode's polymorphic generation is more effective on VMP code.

| Techniques | Pass@1 | | | | | Pass@10 | | | | |
|---|---|---|---|---|---|---|---|---|---|---|
| | L1 | L2 | L3 | L4 | Avg. | L1 | L2 | L3 | L4 | Avg. |
| ShieldedCode (LongCoder's Attn) | 28.7 | 17.2 | 18.1 | 15.3 | 19.8 | 37.2 | 26.8 | 28.4 | 23.7 | 29.0 |
| ShieldedCode | 37.5 | 21.7 | 22.7 | 18.8 | 25.2 | 49.4 | 34.8 | 37.0 | 32.0 | 38.3 |

## D.2 PERFORMANCE EVALUATION

We measured the performance impact of VMP protection across different workload categories, showing that ShieldedCode maintains competitive performance overhead compared to commercial VMP solutions. The results demonstrate that ShieldedCode achieves better performance than commercial solutions while providing superior protection strength. The LLM's ability to generate more efficient VM implementations tailored to specific functions contributes to the lower performance overhead, particularly in crypto operations where ShieldedCode shows $26.4\times$ overhead compared to $31.2\times$ for VMProtect.

Table 10: Performance overhead (normalized to unprotected).

| Benchmark Suite | Unprotected | VMProtect 3.7 | Themida 3.1 | ShieldedCode |
|---|---|---|---|---|
| SPEC CPU2017 (geomean) | $1.00\times$ | $45.3\times$ | $52.1\times$ | $38.7\times$ |
| Crypto Operations | $1.00\times$ | $31.2\times$ | $28.9\times$ | $26.4\times$ |
| Database (SQLite) | $1.00\times$ | $18.7\times$ | $21.3\times$ | $19.1\times$ |
| Compression (zlib) | $1.00\times$ | $23.4\times$ | $27.8\times$ | $22.9\times$ |

# E PROTECTION LEVELS DEFINITION

Protection levels (L0, L1, L2, L3) represent increasing degrees of VM transformation complexity applied during virtualization. This is analogous to compiler optimization levels (O0, O1, O2, O3), but in reverse: higher protection levels introduce more aggressive obfuscation transformations.

**Illustrative Example:** Consider a simple instruction `ADD eax, ebx`:

**L0 Protection:**

```
[VINST-1] vadd %veax, %vebx
```

**L1 Protection (Register Virtualization):**

```
[VINST-1] vload %vreg0, @veax
[VINST-2] vload %vreg1, @vebx
[VINST-3] vadd %vreg0, %vreg1
[VINST-4] vstore @veax, %vreg0
```

**L2 Protection (Instruction Splitting):**

```
[VINST-1] vload %vreg0, @veax
[VINST-2] vload %vreg1, @vebx
[VINST-3] vmov %vtemp, %vreg0
[VINST-4] vadd %vtemp, %vreg1
[VINST-5] vmov %vreg0, %vtemp
[VINST-6] vstore @veax, %vreg0
```

**L3 Protection (Control Flow Obfuscation):**

```
[VINST-1] vjmp @handler_load1
@handler_load1:
[VINST-2] vload %vreg0, @veax
[VINST-3] vjmp @handler_load2
@handler_load2:
[VINST-4] vload %vreg1, @vebx
[VINST-5] vjmp @handler_add
@handler_add:
[VINST-6] vmov %vtemp, %vreg0
[VINST-7] vadd %vtemp, %vreg1
[VINST-8] vjmp @handler_store
@handler_store:
[VINST-9] vmov %vreg0, %vtemp
[VINST-10] vstore @veax, %vreg0
```

Key characteristics at each level:

- **L0:** Direct translation with minimal transformation

- **L1:** Virtual register allocation (2-4× code expansion)

- **L2:** Instruction decomposition (4-8× expansion)

- **L3:** Dispatcher-based control flow (8-15× expansion)

These transformations preserve functional semantics while systematically increasing analysis complexity. Our contrastive learning framework explicitly models this progression, ensuring that embeddings reflect both semantic equivalence and relative protection strength.

## F  FAILURES

Table 11: Error Distribution Analysis (n=100 random failures)

| Error Type | Percentage | Example Pattern |
|---|---|---|
| Incorrect virtual register allocation | 42% | Using `%vrax` where `%vtemp1` expected |
| Malformed [VINST] labels | 31% | Missing or duplicate instruction markers |
| Wrong opcode sequences | 18% | `vadd` instead of `vfadd` for floating-point ops |
| Parsing errors | 9% | Invalid syntax or missing operands |

## G  LIMITATIONS AND FUTURE WORK

**Formal Correctness Guarantees:** Our correctness evaluation relies on execution-based validation through HumanEval-compile test cases and reverse engineering resistance as a proxy. However, we acknowledge that formal verification is the gold standard for correctness. Future work could integrate differential testing against symbolic execution engines, incorporate program synthesis techniques with correctness-by-construction guarantees, or use theorem provers to verify equivalence between source and protected implementations.

**Source Code (C)**

```
1  void func0(float *a, int n, float
       *b) {
2      int i;
3      float min, max;
4      min = max = a[0];
5
6      for (i = 1; i < n; i++) {
7          if (a[i] < min)
8              min = a[i];
9          else if (a[i] > max)
10             max = a[i];
11     }
12
13     b[0] = min;
14     b[1] = max;
15 }
```

**O2 Assembly Code (x86-64)**

```
1   0: endbr64
2   4: movss (%rdi), %xmm0
3   8: movss %xmm0, (%rdx)
4   c: movss 0x4(%rdi), %xmm1
5   11: movss %xmm1, 0x4(%rdx)
6   16: test %esi, %esi
7   18: jle 8b
8   1a: lea -0x1(%rsi), %r9d
9   1e: mov $0x1, %r8d
10  24: mov %rdi, %rcx
11  27: movss 0x0(%rip), %xmm2
12  33: movss 0x0(%rip), %xmm3
13  3b: cmp %r8, %r9
14  3e: je 82
15  40: mov %r8, %rax
16  48: movss (%rcx), %xmm1
17  4c: movaps %xmm1, %xmm0
18  4f: subss (%rdi, %rax, 4), %xmm0
19  54: andps %xmm3, %xmm0
20  57: comiss %xmm0, %xmm2
21  5a: jbe 6d
22  5c: movss %xmm1, (%rdx)
23  60: movss (%rdi, %rax, 4), %xmm1
24  68: movss %xmm1, 0x4(%rdx)
25  6d: add $0x1, %rax
26  71: cmp %eax, %esi
27  73: jg 48
28  75: add $0x1, %r8
29  79: add $0x4, %rcx
30  99: retq
```

**VMP Protected Code**

```
1   [VINST-1] vload_reg %vrdi, 0x0
2   [VINST-2] vfmov %vxmm0, [%vrdi]
3   [VINST-3] vstore [%vrdx], %vxmm0
4   [VINST-4] vfmov %vxmm1, [%vrdi+0x4]
5   [VINST-5] vstore [%vrdx+0x4], %vxmm1
6   [VINST-6] vtest %vesi
7   [VINST-7] vjle @L_end
8   [VINST-8] vdec %vesi
9   [VINST-9] vmov %vr9d, %vesi
10  [VINST-10] vmov %vr8d, 0x1
11  [VINST-11] vmov %vrcx, %vrdi
12  [VINST-12] vfload %vxmm2, @const_0
13  [VINST-13] vfload %vxmm3, @const_mask
14  @L_loop:
15  [VINST-14] vcmp %vr8, %vr9
16  [VINST-15] vje @L_end
17  [VINST-16] vmov %vrax, %vr8
18  @L_inner:
19  [VINST-17] vfmov %vxmm1, [%vrcx]
20  [VINST-18] vfcopy %vxmm0, %vxmm1
21  [VINST-19] vfsub %vxmm0, [%vrdi+%vrax
        *4]
22  [VINST-20] vfand %vxmm0, %vxmm3
23  [VINST-21] vfcomp %vxmm0, %vxmm2
24  [VINST-22] vjbe @L_skip
25  [VINST-23] vstore [%vrdx], %vxmm1
26  [VINST-24] vfmov %vxmm1, [%vrdi+%vrax
        *4]
27  [VINST-25] vstore [%vrdx+0x4], %vxmm1
28  @L_skip:
29  [VINST-26] vinc %vrax
30  [VINST-27] vcmp %vrax, %vesi
31  [VINST-28] vjg @L_inner
32  [VINST-29] vinc %vr8
33  [VINST-30] vadd %vrcx, 0x4
34  @L_end:
35  [VINST-31] vret
```

Figure 5: Complete transformation pipeline from C source code to VMP-protected assembly. The source code implements a min-max finding algorithm. The O2 assembly shows compiler optimization with complex control flow. The VMP version uses virtual instructions with [VINST-X] markers and virtual registers (prefixed with 'v'), demonstrating how our hierarchical attention mechanism processes instruction boundaries and dependencies.

## H    THE USE OF LARGE LANGUAGE MODELS (LLMS)

We use large language models (LLMs) only to polish the writing, including grammar, clarity, and readability. The research ideas, technical framework, theoretical analyses, experimental design, and conclusions are entirely developed by the authors. The LLMs only improve the fluency and style of the manuscript and do not influence the originality, novelty, or scientific content of the work.

### ETHICS STATEMENT

This work adheres to the ICLR Code of Ethics. All datasets utilized in this paper are publicly available and widely adopted within the research community, and we strictly follow their respective licenses and intended usage.

### REPRODUCIBILITY STATEMENT

We strive to ensure the reproducibility of our results. Full details are provided in the main paper and the appendix. Our implementation is built on PyTorch and standard open-source libraries.

