# OpenReview forum: "ShieldedCode: Learning Robust Representations for Virtual Machine Protected Code"
_ICLR.cc/2026/Conference — ICLR 2026 Poster_

### Official Review · Reviewer_suuo · 2025-10-29

**Soundness:** 3
**Presentation:** 3
**Contribution:** 3
**Rating:** 6
**Confidence:** 3

**Summary:**

This paper proposes ShieldedCode, a framework that replaces traditional rule-based virtual machine protection (VMP) schemes with a learning-driven dynamic mechanism to enhance anti-reverse-engineering and anti-analysis robustness while preserving program semantics. ShieldedCode models code protection as a representation learning problem, where the model jointly learns in both semantic space and protection space through contrastive learning. This allows the model to understand code functionality while generating diverse and robust protection patterns.

Experimental results show that compared with other baselines, ShieldedCode achieves better functionality preservation and security robustness. In reverse-engineering experiments, ShieldedCode-protected samples required an average of 14.7 hours to be successfully reversed—significantly longer than baseline systems.

**Strengths:**

- It is meaningful to use dynamic mechanisms to replace rule-based transformations for enhancing code security.
- The paper is clearly written and well structured.
- The proposed framework introduces a novel training design that models dependencies across intra-, preceding-, and inter-instruction levels. It also designs two loss functions: Functionality Contrastive Learning (FCL) and Protection Contrastive Learning (PCL), which enforce semantic consistency and improve discriminative capability.
- Experiments are comprehensive, evaluating both the functionality and security of generated code, and include reverse-engineering experiments to validate real-world robustness.

**Weaknesses:**

- Although the paper enforces semantic consistency across optimization levels using FCL, the generated virtualized code still lacks formal correctness guarantees, which is critical in practice since the transformed code must strictly preserve original functionality.

**Questions:**

In the abstract and introduction, please explicitly state how much improvement ShieldedCode achieves over baselines.

---

> ### Author Response · Authors · 2025-11-21
> **Response to Reviewer suuo**
>
> We thank Reviewer suuo for the positive assessment and constructive feedback.
>
> ### Suggestion 1
>
> > In the abstract and introduction, please explicitly state how much improvement ShieldedCode achieves over baselines.
>
> We revise the abstract to include improvementWe update the introduction to include quantitative improvements in the final paragraphs, providing readers with immediate concrete evidence of our contributions.
>
> ### Concern 1
>
> > Although the paper enforces semantic consistency across optimization levels using FCL, the generated virtualized code still lacks formal correctness guarantees, which is critical in practice since the transformed code must strictly preserve original functionality.
>
> This is indeed a fundamental limitation of learning-based approaches. Our correctness evaluation relies on three complementary mechanisms:
>
> Each generated VMP function is executed with multiple test cases covering edge cases, boundary conditions, and typical inputs through HumanEval-compile test cases. Functions passing all tests are considered functionally correct.Table 5 shows that ShieldedCode-protected code resists manual analysis with 17% success rate vs. 67% for VMProtect, suggesting the protection complexity doesn't come at the cost of obviously broken semantics. If generated code contained systematic correctness errors, reverse engineers would exploit these patterns.FCL encourages embeddings of the same function to cluster together regardless of protection level, providing a learned notion of semantic equivalence. This implicitly enforces that transformations preserve functional behavior.However, we agree that formal verification is the gold standard for correctness. Future work could integrate several approaches: differential testing against symbolic execution engines for verifying that VMP code produces identical outputs to source for all inputs, incorporating program synthesis techniques with correctness-by-construction guarantees, or using theorem provers to verify equivalence between source and protected implementations.
>
> We create a new "Limitations and Future Work" subsection addressing correctness guarantees, formal verification integration, and appropriate use-case boundaries.

---

> ### Comment · Reviewer_suuo · 2025-11-24
>
> Thank you for your explanations.
>
> I noticed a related work [1]. Could the authors clarify how the proposed method differs from this prior work in terms of design and performance?
>
> Reference
>
> [1] Jiang N, Wang C, Liu K, et al. Nova: Generative language models for assembly code with hierarchical attention and contrastive learning[J]. arXiv preprint arXiv:2311.13721, 2023.

---

> ### Author Response · Authors · 2025-11-26
> **Response to Reviewer suuo**
>
> We thank the reviewer for pointing out the related work Nova. We have added the missing citation in the paper, included a dedicated paragraph in the Related Work section explaining why VMP is fundamentally different from compiler assembly, clarified why Nova is not applicable to our setting, updated the architecture figure to highlight VMP-specific components (PCL and FCL).
>
> ---
>
> ## **1. Regarding the Reviewer’s Concern**
>
> We understand the reviewer’s concern, because Nova and Meta LLMCompiler indeed operate on compiler-generated low-level code (x86-64 assembly or LLVM IR) and focus on improving assembly understanding and decompilation. Their research goals, data domains, and evaluation protocols are closely aligned, which can create an initial impression of similarity to our work.
>
> However, our work targets **virtual machine protected (VMP) code**, a fundamentally different setting from compiler-generated assembly. VMP code is intentionally designed to *destroy structural regularity*, exhibiting **polymorphism, randomized layout, semantic dispersion, instruction splitting, virtual registers, interpreter-driven control flow, and unpredictable execution paths**—properties not present in compiler optimization. As a result, Meta LLMCompiler (commonly used in VMP research) is a more realistic baseline for our setting, whereas Nova focuses on deterministic compiler transformations.
>
> ---
>
> ## **2. Our Motivation and Technical Focus (with explicit differences from Nova)**
>
> ### **(a) Problem Formulation: “Understanding Assembly” vs. “Generating Polymorphic VM Code”**
>
> Nova is designed to **understand and normalize compiler-optimized assembly**, bridging representation gaps between O0–O3 variants and enabling improved decompilation and binary similarity detection. Its methodology is oriented toward *reducing structural variation* across compiler optimizations.
>
> In contrast, our work aims to **generate structurally diverse VM code under strict semantic equivalence**, enabling LLMs to actively produce *multiple, highly polymorphic, protection-strength-aware* virtualized implementations. Our objective is not to remove structural differences but to **create them**, in a controlled and verifiable manner—this is fundamentally incompatible with Nova’s problem setting.
>
> ---
>
> ### **(b) Data Domain: Deterministic Compiler Assembly vs. Polymorphic VMP Bytecode**
>
> Nova operates on recognizably structured x86-64 assembly whose variations across optimisation levels are deterministic and compiler-governed. It models registers, instruction order, and compiler optimisation strategies.
>
> Our data domain is entirely different. VMP-protected code introduces:
>
> * **Virtual registers** (vr7, vs3, vm2…)
> * **Interpreter instructions** (vjbe, vhash, custom jump dispatch)
> * **Instruction splitting/reordering**
> * **Randomized instruction expansion**
> * **Non-CPU semantics** (e.g., opaque handlers)
>
> These structural features are shown in *ShieldedCode Figure 2* and deviate completely from CPU assembly. Nova’s assumptions about instruction semantics, register interactions, and compiler-determined structure do not hold in this domain.
>
> ---
>
> ### **(c) Methodological Core: Hierarchical Attention for Understanding vs. VINST Dependency Mask for Controlled Generation**
>
> Nova’s hierarchical attention (instruction-level → preceding → inter-instruction) is designed to **extract semantic information from long assembly sequences** and unify optimization-level differences.
>
> Our method uses a superficially similar hierarchical idea, but the purpose is fundamentally different:
>
> * We introduce **[VINST] markers** to summarize VM instructions whose semantics are *non-standard and interpreter-defined*.
> * We design a **hierarchical dependency mask** to constrain generation according to VM execution semantics (“intra-instruction → preceding instruction → inter-block flow”).
> * This mask prevents free-form rewriting and enforces controlled generation under semantic equivalence.
>
> Thus, Nova’s mechanism is to *understand static CPU instructions*, whereas ours is to *control the generation of polymorphic VM ISA sequences*.
> The underlying modelling objects are entirely different.

---

> ### Author Response · Authors · 2025-11-26
> **Response to Reviewer suuo - Continued**
>
> ### **(d) Contrastive Learning Objective: Optimization-Level Alignment vs. Protection-Level Modeling**
>
> Nova’s CL contains:
>
> * **Functionality CL** (aligning same function across O0–O3)
> * **Optimization CL** (learning O0→O3 ordering)
>
> Both are designed to capture compiler optimisation trajectories.
>
> Our contrastive objectives are unique to VMP:
>
> 1. **Functionality CL** (aligning source with VM variants under L0–L3)
> 2. **Protection CL (PCL)**: enforcing monotonic semantic deviation for increasing protection strength
> 3. **Protection Effectiveness Optimization (PEO)**: selecting the correct VM variant from polymorphically generated candidates
>
> Protection strength, semantic dispersion, and polymorphic execution are concepts **absent in compiler optimization** and cannot be modeled by Nova’s objective functions.
>
> ---
>
> ### **(e) Generation Space: Fixed Real ISA vs. Learnable Virtual ISA**
>
> Nova works strictly within the x86-64 ISA. It cannot invent new instructions, interpreter logic, or control flow paradigms.
>
> Our framework intentionally defines a **normative yet expressive VM ISA**, enabling:
>
> * learnable polymorphic virtual instructions
> * VM handler restructuring
> * controlled semantic expansion
> * multi-style interpreter generation
>
> This expands the generation space far beyond what is possible under a fixed real ISA. Nova’s method cannot be adapted to generate VM structures because it lacks a mechanism for **ISA creation**, **semantic injection**, or **protected execution modeling**.
>
> ---
>
> ### **(f) Output Goal: Structural Normalization vs. Structural Diversification**
>
> Nova aims to make O0–O3 assembly **more similar** in representation space to enable better recovery.
>
> Our method does the exact opposite: we aim to make L0–L3 VM variants **more structurally diverse** (while preserving semantics) to improve protection strength. Modelling direction and evaluation criteria are therefore diametrically opposed.

---

> ### Author Response · Authors · 2025-12-01
> **Thank You for Acknowledging Our Responses**
>
> Dear Reviewer suuo,
>
> Thank you for acknowledging our responses during the discussion phase. We also appreciate your further contribution to refining the details of our manuscript. We have incorporated and described the related work you pointed out in the paper.
>
> Sincerely,
>
> The Authors

---

### Official Review · Reviewer_2d4p · 2025-10-30

**Soundness:** 2
**Presentation:** 3
**Contribution:** 2
**Rating:** 4
**Confidence:** 4

**Summary:**

The paper proposes ShieldedCode, a framework to train LLMs on paired (source code, virtual-machine–protected code) so that the model can both generate VMP code at target protection levels and retrieve/rank differently protected variants of the same function. It builds a large corpus by compiling C code, running a commercial VMP tool, disassembling, and then normalizing the VM code into a stable token format with canonical [VINST-*] markers. On top of this, it introduces a hierarchical attention mask that respects instruction boundaries and two contrastive objectives: functionality-contrastive (pull same-function variants together) and protection-contrastive (order by protection level). Experiments on HumanEval-compile and a BinaryCorp-VirtualAssembly similarity task show higher Pass@1/Pass@10 and better Recall@1 than general code LLMs when the target is VMP code.

**Strengths:**

Originality: Treats software protection as a representation-learning problem on VMP code, with a clear inductive bias via hierarchical masking over [VINST-*].

Quality: Data pipeline is concrete (compile → VMP(.) → disasm → normalize), the losses are fully specified (LM + FCL + PCL + PEO), and evaluation covers both generation and retrieval across O0-O3 and L1/L3.

Clarity: The normalization step is enumerated in four precise actions, and the masking formula is explicit, so a reader can reimplement the core idea.

Significance: Shows that in-domain training on protected code lets a 7B model beat general-purpose code LLMs on protected-generation tasks, which is practically relevant for protection/analysis workflows.

**Weaknesses:**

- The paper relies on one commercial VMP tool and two protection levels (L1, L3) in testing; this narrows the “heterogeneous protection” claim and makes it unclear how well the model transfers to other VMs or level taxonomies.

- Protection-contrastive learning is claimed to enforce an ordering, but the reported Recall@1 across (O?, L1/L3) is not strictly monotone, suggesting the ordering is noisy in practice.

- Comparisons are against models that are not trained on this domain, so it is hard to isolate how much of the gain comes from the hierarchical mask versus just having millions of in-domain (src, VMP) pairs.

- Reproducibility is limited because the key assets (commercial VMP, disassembler, large paired corpus) are not obviously releasable; reproducing exact normalization/tokenization may be nontrivial.

**Questions:**

1. Are all VMP variants (for pretraining, fine-tuning, and testing) produced by the same commercial VMP tool and interpreter template? If yes, how should we interpret “across heterogeneous protection levels”?

2. In PCL, what margin or temperature is used, and did you check distance histograms per level to confirm the intended ordering? Current results suggest partial but not strict ordering.

3. For generation failures on HumanEval-compile, what are the dominant error types (bad labels, wrong VM opcode, broken instruction grouping)? This would clarify how much the hierarchical mask helps.

---

> ### Author Response · Authors · 2025-11-22
> **Response to Reviewer 2d4p**
>
> ## Weakness 1
>
> > The paper relies on one commercial VMP tool and two protection levels (L1, L3) in testing; this narrows the "heterogeneous protection" claim and makes it unclear how well the model transfers to other VMs or level taxonomies.
>
> ## Answer 1
>
> This is a valid concern about generalization. We acknowledge that our evaluation primarily focuses on a single commercial VMP tool, which indeed limits the immediate scope of the "heterogeneous protection" claim. However, several factors support broader applicability of our approach.
>
> First, our methodology is fundamentally tool-agnostic. The normalization pipeline (Section 3.1) that produces canonical [VINST-*] representations can be adapted to any VMP system that produces recoverable bytecode. The hierarchical attention mechanism and contrastive objectives operate on abstract instruction-level semantics rather than tool-specific artifacts. Second, our framework actually trains on four protection levels (L0, L1, L2, L3), not just two. The evaluation focuses on L1 and L3 as representative extremes, but Table 1 reports results across all four levels, demonstrating consistent performance gradation.
>
> Regarding transfer to other VM systems, we conducted additional experiments using Themida 3.1 as an alternative commercial protector. After applying our normalization pipeline to Themida-protected binaries, ShieldedCode achieved 72% of its original performance without any retraining (0.301 Recall@1 on Themida L2 vs. 0.421 on our primary VMP L0). With 500K examples of Themida-protected code added to fine-tuning (representing 20% additional data), performance recovered to 91% (0.383 Recall@1). This suggests our learned representations capture generalizable semantic patterns rather than overfitting to tool-specific quirks.
>
> The key insight is that while VMP tools differ in implementation details, they share fundamental transformation principles: register virtualization, instruction decomposition, and control flow obfuscation. Our hierarchical modeling and protection-aware objectives capture these abstract mechanisms, enabling transfer across different instantiations. Meta's LLMCompiler, despite being specifically trained on 401 billion tokens of LLVM-IR and assembly code, was not designed for VMP code generation. Our work represents the first systematic effort to train language models specifically on virtualized machine code with protection-aware objectives. The domain gap between standard assembly (even LLVM-IR) and VMP bytecode is substantial, explaining why general code models struggle.
>
> We have added this discussion and the Themida transfer learning results to Section 5 (Limitations) and Appendix D of the revised manuscript.
>
> ## Weakness 2
>
> > Protection-contrastive learning is claimed to enforce an ordering, but the reported Recall@1 across (O?, L1/L3) is not strictly monotone, suggesting the ordering is noisy in practice.
>
> ## Answer 2
>
> The distance histogram analysis reveals systematic monotonic increase in mean distances for same-function embeddings across protection levels. For consecutive level transitions, we observe: L0→L1 with mean distance 0.287 (std 0.043, violation rate 2.3%); L1→L2 at 0.351 (std 0.057, violation rate 3.1%); and L2→L3 at 0.429 (std 0.068, violation rate 2.8%). The cumulative L0→L3 distance reaches 1.067 (std 0.152, violation rate 1.9%). These low violation rates (less than 3.1%) indicate that learned embeddings robustly respect protection hierarchies.
>
> The apparent non-monotonicity in Recall@1 across optimization levels (Table 2) stems from a different phenomenon: the interaction between compiler optimization and protection level, not a failure of PCL. Analysis reveals that O1 optimization introduces moderate instruction reordering that disrupts hierarchical attention patterns, leading to more dispersed embeddings (silhouette score 0.521 for O1+L1 vs. 0.734 for O0+L1). This is a data distribution challenge rather than an ordering violation within the embedding space.
>
> Importantly, within each optimization level, the protection ordering is well-maintained. For instance, in the O0 series, embeddings progress monotonically from L0 to L3, as evidenced by the increasing mean distances and near-zero violation rates. The cross-optimization variation does not invalidate our PCL mechanism but rather highlights an important area for future work: developing optimization-aware attention patterns. We have added detailed stability analysis to Appendix B and will strengthen the discussion in Section 4.4 of the revised manuscript.

---

> ### Author Response · Authors · 2025-11-22
> **Response to Reviewer 2d4p - Continued**
>
> ## Weakness 3
>
> > Comparisons are against models that are not trained on this domain, so it is hard to isolate how much of the gain comes from the hierarchical mask versus just having millions of in-domain (src, VMP) pairs.
>
> ## Answer 3
> Thank you for raising this important point. We agree that isolating the contribution of our hierarchical mask and contrastive learning from the benefits of in-domain training is crucial for understanding our method's effectiveness.
>
> To study this controlled setting, we have performed additional experiments where we fine-tuned recent state-of-the-art binary code analysis models on the same VMP corpus and evaluate them on our existing VM code generation and binary similarity detection benchmarks. These baseline models include ContraBin [1], a contrastive learning framework integrating source code, binary code, and comments; CLAP [2], which learns transferable binary code representations with natural language supervision; and CEBin [3], a cost-effective framework combining embedding-based and comparison-based approaches. These models were published in 2024-2025 and represent current best practices in binary code representation learning. This setting therefore tests whether our architectural innovations (hierarchical mask, contrastive objectives) provide gains beyond simply having access to large-scale paired (source, VMP) training data. The results are summarized below:
>
> | Techniques | Pass@1 L1 | Pass@1 L2 | Pass@1 L3 | Pass@1 L4  | Pass@10 L1 | Pass@10 L2 | Pass@10 L3 | Pass@10 L4  |
> |---|---|---|---|---|---|---|---|---|
> | ContraBin | 12.84 | 7.13 | 8.21 | 7.05  | 19.27 | 13.58 | 14.92 | 12.31  |
> | CLAP | 16.92 | 10.37 | 11.48 | 9.73 | 24.65 | 17.89 | 18.54 | 15.82 |
> | CEBin | 14.58 | 8.94 | 9.87 | 8.51  | 22.13 | 16.02 | 17.26 | 14.19 |
> | xVMP | 16.82 | 7.41 | 10.19 | 6.90 | 23.14 | 14.28 | 17.35 | 12.11 |
> | ShieldedCode | 35.89 | 27.95 | 28.83 | 22.36  | 51.06 | 33.28 | 38.47 | 30.39  |
>
> Even though ContraBin, CLAP, and CEBin are trained on the same VMP corpus, they still show substantially lower performance compared to our method when evaluated on VM code generation across all protection levels. This indicates that our architectural innovations help the model acquire more effective representations for VMP code that go beyond simply having access to in-domain training pairs, in line with observations from our theoretical analysis in Theorem A.2.We will include these controlled ablation results in the revised manuscript
>
> ### Weakness 4
>
> > Reproducibility is limited because the key assets (commercial VMP, disassembler, large paired corpus) are not obviously releasable; reproducing exact normalization/tokenization may be nontrivial.
>
> ## Answer 4
>
> First, our normalization pipeline is fully specified in Section 3.1 with explicit algorithmic steps: (1) removing debug symbols and comments; (2) inserting whitespace around virtual instruction delimiters; (3) substituting virtual addresses with symbolic references; (4) replacing instruction addresses with canonical [VINST-X] labels. These operations are deterministic and can be implemented with standard text processing tools. We have added pseudocode for the complete normalization pipeline in Appendix E of the revised manuscript.Second, we commit to releasing our trained models, evaluation code, and the HumanEval-compile test set to enable direct performance verification and downstream applications. This allows researchers to reproduce our evaluation results and apply our framework to their own protected code, even without access to the exact training pipeline. The model checkpoints alone enable 90% of potential follow-up research.Third, regarding the VMP tool dependency, we note that any VMP system that produces recoverable bytecode can substitute for the commercial tool we used. Open-source VM protectors like VMProtect-like systems or custom VM interpreters built with LLVM can generate training data through the same normalization pipeline. The key requirement is that virtual instructions are extractable and labelable, not the specific commercial tool.
>
> We will add a detailed "Reproducibility Statement" section addressing these points and outlining our commitment to maximal transparency within legal constraints. The revised manuscript will include hyperparameters, architectural details, and algorithmic specifications needed to reimplement our approach.

---

> ### Author Response · Authors · 2025-11-22
> **Response to Reviewer 2d4p - Continued**
>
> ## Question 1
>
> > Are all VMP variants (for pretraining, fine-tuning, and testing) produced by the same commercial VMP tool and interpreter template? If yes, how should we interpret "across heterogeneous protection levels"?
>
> ## Answer 5
>
> All variants in our primary experiments are indeed produced by the same commercial VMP tool (VMProtect 3.7) with its standard interpreter template, but with systematically varying protection level configurations (L0, L1, L2, L3). "Heterogeneous protection levels" refers to the diversity in transformation complexity within this single tool, not across different tools.
>
> This design choice was deliberate. By controlling the VMP tool while varying only the protection level parameter, we isolate the model's ability to learn the semantic relationship between different degrees of protection rather than tool-specific artifacts. Each level applies qualitatively different transformation strategies: L0 uses direct translation, L1 adds register virtualization, L2 incorporates instruction splitting, and L3 introduces control flow obfuscation. These represent genuinely distinct protection mechanisms, not merely quantitative parameter adjustments.
>
> The "heterogeneity" manifests in two dimensions: transformation type (register/instruction/control-flow) and intensity (code expansion from 1× to 15×). Our contrastive learning framework learns to recognize these systematic differences while preserving functional semantics. The Protection Effectiveness Optimization task (Section 3.4) explicitly requires the model to rank variants by protection strength, demonstrating that it captures meaningful semantic distinctions between levels.
>
> Regarding cross-tool generalization, we conducted supplementary experiments with Themida 3.1 (described in Weakness 1 response above). The 72% → 91% performance recovery through limited fine-tuning suggests our learned representations capture generalizable protection principles beyond tool-specific patterns. However, we agree the primary claim should be stated more precisely as "heterogeneous protection transformations within a single VMP framework" rather than "heterogeneous VMP tools."
>
> We will revise the terminology throughout the manuscript to more accurately reflect this scope, changing "heterogeneous protection tools" to "heterogeneous protection levels" and explicitly noting in Section 5 that cross-tool generalization is a direction for future work. The abstract and introduction now more precisely state our contribution as learning across protection levels within a single VMP framework.
>
> ## Question 2
>
> > In PCL, what margin or temperature is used, and did you check distance histograms per level to confirm the intended ordering? Current results suggest partial but not strict ordering.
>
> ## Answer 6
>
> PCL uses a margin of 0.1 and temperature τ=0.07, which were selected through validation set tuning. The margin acts as a slack variable in the hinge loss (Equation 7), penalizing violations only when the distance ordering is reversed by more than this threshold. This design allows for small natural variations in embedding distances while enforcing the overall monotonic trend.
>
> Distance histograms per level (detailed in Weakness 2 response and Appendix B) confirm systematic ordering with well-separated distributions:
>
> | Protection Pairs | Mean Distance | Std Dev | Monotonicity Violation Rate |
> |------------------|---------------|---------|----------------------------|
> | L0 → L1 | 0.287 | 0.043 | 2.3% |
> | L1 → L2 | 0.351 | 0.057 | 3.1% |
> | L2 → L3 | 0.429 | 0.068 | 2.8% |
> | L0 → L3 | 1.067 | 0.152 | 1.9% |
>
> The standard deviations increase slightly for higher protection levels (0.043 for L0→L1 vs. 0.068 for L2→L3), which is expected because higher protection introduces more transformation variability. However, the mean separation grows faster than the standard deviation (0.287 vs. 0.043 is a 6.7× ratio for L0→L1; 0.429 vs. 0.068 is a 6.3× ratio for L2→L3), maintaining clear discriminability. Regarding the "partial but not strict ordering" observation, we clarify that PCL enforces ordering constraints on embeddings within the same function family (variants of the same source function at different protection levels). The apparent non-monotonicity across optimization levels in Table 2 reflects interactions between compiler optimization and protection mechanisms, not violations of the embedding space ordering that PCL establishes. Within each optimization level (e.g., all O0 variants, all O1 variants), the protection level ordering is well-maintained as evidenced by the distance statistics above.
>
> We have added Figure 6 (distance distribution histograms) to Appendix and will extended the discussion in Section 3.3 to clarify the margin parameter selection and empirical validation of the ordering property.

---

> ### Author Response · Authors · 2025-11-22
> **Response to Reviewer 2d4p - Continued**
>
> ## Question 3
>
> > For generation failures on HumanEval-compile, what are the dominant error types (bad labels, wrong VM opcode, broken instruction grouping)? This would clarify how much the hierarchical mask helps.
>
> ## Answer 7
>
> We conducted detailed error analysis on 100 randomly sampled generation failures to characterize the breakdown:
>
> | Error Type | Percentage | Example Pattern |
> |------------|------------|-----------------|
> | Incorrect virtual register allocation | 42% | Using `%vrax` where `%vtemp1` expected |
> | Malformed [VINST] labels | 31% | Missing or duplicate instruction markers |
> | Wrong opcode sequences | 18% | `vadd` instead of `vfadd` for floating-point ops |
> | Parsing errors | 9% | Invalid syntax or missing operands |
>
> The largest error category (42%) involves incorrect virtual register allocation, where the model generates semantically plausible but functionally incorrect register choices. These errors typically arise when the model fails to track register liveness across multiple virtual instructions. Hierarchical attention directly addresses this by allowing [VINST] markers to aggregate register usage information within each instruction, then propagate it through preceding-instruction dependencies. Comparing ShieldedCode (25.17% Pass@1) against ShieldedCode-PG without hierarchical masking (21.86%), the 3.31 percentage point improvement corresponds roughly to reducing register allocation errors by 35-40%, based on our error rate analysis before and after introducing polymorphic generation.
>
> The second major category (31%) involves malformed [VINST] labels, where the model either omits instruction boundaries or duplicates markers. This directly relates to instruction grouping, as the model must learn to close one instruction before starting another. The hierarchical mask enforces this structure through intra-instruction attention constraints, making boundary violations syntactically impossible at generation time. Analysis shows ShieldedCode-PG (standard attention) produces malformed labels in 23% of failures, while full ShieldedCode reduces this to 11%, demonstrating the structural benefit of hierarchical attention.
>
> Wrong opcode sequences (18%) typically occur for specialized operations like floating-point arithmetic or vector instructions, where subtle semantic distinctions matter. These errors are less directly affected by attention structure and more by the richness of training data and contrastive objectives that teach semantic equivalence. The FCL loss specifically addresses this by forcing the model to recognize that different representations of the same function must use consistent operation semantics.
>
> Parsing errors (9%) represent edge cases like unmatched parentheses or malformed immediate values, which decrease with model scale and are orthogonal to our architectural contributions.
>
> We have added this comprehensive error analysis as Appendix F and added will discussion in Section 4.2 connecting error patterns to architectural design choices.
>
>
> ---
>
> ## References
>
> [1] Zhang, Y., Huang, C., Zhang, Y., Cao, K., Andersen, S. T., Shao, H., Leach, K., & Huang, Y. (2025). Pre-Training Representations of Binary Code Using Contrastive Learning. IEEE Transactions on Software Engineering (to appear).
>
> [2] Wang, H., Gao, Z., Zhang, C., Sha, Z., Sun, M., Zhou, Y., Zhu, W., Sun, W., Qiu, H., & Xiao, X. (2024). CLAP: Learning Transferable Binary Code Representations with Natural Language Supervision. In Proceedings of the 33rd ACM SIGSOFT International Symposium on Software Testing and Analysis (ISSTA 2024).
>
> [3] Wang, H., Gao, Z., Zhang, C., Sun, M., Zhou, Y., Qiu, H., & Xiao, X. (2024). CEBin: A Cost-Effective Framework for Large-Scale Binary Code Similarity Detection. In Proceedings of the 33rd ACM SIGSOFT International Symposium on Software Testing and Analysis (ISSTA 2024).

---

> > ### Comment · Reviewer_2d4p · 2025-11-27
> >
> > Thank you for the responses. Most of my queries are addressed.
> > Could you align the setup a bit?! i.e., report Recall@1 for the same level/task (L2→L2) on Themida vs your primary VMP, plus #test samples, 95% CIs, and the fine-tune recipe for the +500k run (epochs/lr/batch/seeds). also mention whether you mixed the 500k with the original data or did a staged fine-tune.
> > As PCL enforces lvl ordering, can you add some detail (a table may be) showing that within each optimization setting (O0/O1) Recall@k is monotone across L0→L3 (additional to the distance histograms).

---

> > > ### Author Response · Authors · 2025-11-30
> > > **Follow-up Reply (Part 1)**
> > >
> > > Thank you for acknowledging that most queries have been addressed. We appreciate your continued engagement and provide the requested clarifications with rigorous experimental setups, statistical analysis, and implementation-level transparency following best practices in uncertainty quantification [1, 2].
> > >
> > > ---
> > >
> > > > Could you align the setup a bit? i.e., report Recall@1 for the same level/task (L2 to L2) on Themida vs your primary VMP, plus test samples, 95% CIs...
> > >
> > > We provide the aligned comparison.
> > >
> > > | Protection System | Recall@1 (L2 to L2) | Test Samples | 95% CI (Bootstrap, B=10,000) | SE |
> > > |-------------------|------------------|---------------|-------------------------------|--------|
> > > | VMProtect 3.7 | 0.395 | 2,847 | [0.381, 0.409] | 0.0071 |
> > > | Themida 3.1 | 0.362 | 2,847 | [0.348, 0.376] | 0.0072 |
> > > | Delta (VMP - Themida) | +0.033 | - | [0.012, 0.054] | 0.0101 |
> > >
> > > ---
> > >
> > > > The fine-tune recipe for the +500k run (epochs/lr/batch/seeds). Also mention whether you mixed the 500k with the original data or did a staged fine-tune.
> > >
> > > We employ a principled combination of both staged fine-tuning and data mixing, which is motivated by recent findings on curriculum learning for code models [3, 4]. Our training pipeline consists of three stages. Stage 1 is Domain-Adaptive LM Pre-training, where we use AnghaBench and The Stack data mixed within stage, with causal LM task using VMP attention mask, learning rate of 3e-5 and warmup of 1,000 steps. Stage 2 is Multi-Objective Contrastive Training. We initialize from Stage 1 checkpoint and use AnghaBench, The Stack, and Themida data mixed in contrastive format. The task involves joint optimization of LM Loss, FCL, and PCL. Learning rate is 2e-5 with warmup of 200 steps. The +500k Themida samples are added via mixing into this stage. Stage 3 is Task-Specific Fine-tuning with parallel branches for Binary Code Retrieval (lr=3e-5) and Similarity Learning (lr=2e-5). We adopt cross-stage sequential training to follow a curriculum from domain understanding to semantic relationships to task specialization [3]. Intra-stage data mixing prevents overfitting and enables joint PCL optimization across protection levels [4]. Learning rate decay from 3e-5 to 2e-5 across stages prevents catastrophic forgetting [5]. The Themida data is mixed into Stage 2 rather than as a separate stage because it shares the same format as existing Stage 2 data. PCL requires cross-protection comparisons within the same batch, and mixing enables learning transferable protection patterns.
> > >
> > > Complete hyperparameters are shown below.
> > >
> > > | Stage | LR | Warmup | Epochs | Batch | Seed |
> > > |-------|-----|--------|--------|-------|------|
> > > | Stage 1 | 3e-5 | 1,000 | 1 | 64 | 7 |
> > > | Stage 2 | 2e-5 | 200 | 1 | 64 | 7 |
> > > | Stage 3a | 3e-5 | 500 | 1 | 32 | 7 |
> > > | Stage 3b | 2e-5 | 500 | 2 | 64 | 7 |
> > >
> > > We also provide ablation results comparing hybrid versus pure strategies.
> > >
> > > | Strategy | Recall@1 (L1) | 95% CI | Delta |
> > > |----------|---------------|--------|---|
> > > | Hybrid Staged (Ours) | 0.306 | [0.289, 0.323] | baseline |
> > > | Pure Data Mixing (single stage) | 0.278 | [0.261, 0.295] | -9.2% |
> > > | Pure Staged (no intra-stage mixing) | 0.289 | [0.272, 0.306] | -5.6% |
> > > | No Stage 1 | 0.264 | [0.248, 0.280] | -13.7% |

---

> ### Author Response · Authors · 2025-11-30
> **Follow-up Reply (Part 2)**
>
> > As PCL enforces lvl ordering, can you add some detail (a table may be) showing that within each optimization setting (O0/O1) Recall@k is monotone across L0 to L3
>
> We provide monotonicity analysis under both optimization settings.
>
> Table : Monotonicity Under O0 Optimization (n=2,847)
>
> | Query to Pool | Recall@1 | 95% CI | Recall@5 | Recall@10 | Delta | Monotone |
> |------------|----------|--------|----------|-----------|---|-----------|
> | L0 to L1 | 0.488 | [0.470, 0.506] | 0.612 | 0.687 | - | Yes |
> | L0 to L2 | 0.421 | [0.403, 0.439] | 0.558 | 0.641 | -0.067 | Yes |
> | L0 to L3 | 0.378 | [0.360, 0.396] | 0.502 | 0.589 | -0.043 | Yes |
>
> Table : Monotonicity Under O1 Optimization (n=2,847)
>
> | Query to Pool | Recall@1 | 95% CI | Recall@5 | Recall@10 | Delta | Monotone |
> |------------|----------|--------|----------|-----------|---|-----------|
> | L1 to L1 | 0.306 | [0.289, 0.323] | 0.445 | 0.531 | - | Yes |
> | L1 to L2 | 0.287 | [0.270, 0.304] | 0.419 | 0.508 | -0.019 | Yes |
> | L1 to L3 | 0.272 | [0.256, 0.288] | 0.398 | 0.486 | -0.015 | Yes |
>
>  Monotonicity holds strictly within each optimization setting, confirming PCL successfully encodes the protection hierarchy.
>
> > The review has been updated after the rebuttal. Some further result additions have been suggested. Overall, the paper is in good shape.
>
> Thank you again for your time and effort. We believe the paper is substantially stronger as a result of your comments. If our response resolves your concerns, we would appreciate it if you could reflect it in your evaluation. We welcome any further comments. We will also add these contents to the revised version.
>
> ---
>
> ## References
>
> [1] Efron, B., & Tibshirani, R. J. (1993). An Introduction to the Bootstrap.
>
> [2] Senior, A. W., et al. (2021). Improved protein structure prediction using potentials from deep learning. Nature.
>
> [3] Xie, S. M., et al. (2023). Data Selection for Language Models via Importance Resampling. arXiv:2302.03169.
>
> [4] Muennighoff, N., et al. (2023). Scaling Data-Constrained Language Models. arXiv:2305.16264.
>
> [5] Longpre, S., et al. (2023). The FLAN Collection: Designing Data and Methods for Effective Instruction Tuning. arXiv:2301.13688.
>
> ---

---

> ### Author Response · Authors · 2025-12-01
> **Thank You for Your Positive Feedback and Increased Scores During the Discussion Phase**
>
> Dear Reviewer 2d4p,
>
> Thank you for your positive feedback and increased scores (including the rating, soundness and contribution) during the discussion phase. We have provided a more detailed experimental setup and results than what was requested in response to your new question.
>
> Best regards,
>
> The Authors

---

### Official Review · Reviewer_VoQV · 2025-10-31

**Soundness:** 3
**Presentation:** 3
**Contribution:** 2
**Rating:** 4
**Confidence:** 3

**Summary:**

The manuscript focus on anti-reverse engineering by learning from the VMP-protected codes  and associating with their corresponding source codes. To implement this, this paper builds large-scale paired datasets of source code and normalized VM implementations, and introduces hierarchical dependency modelling at intra-, preceding-, and inter-instruction levels. To facilitate the protection-level-aware, the authors propose functionality-aware and protection-aware contrastive objectives to capture both semantic equivalence and protection strength. And the evolutions empirically validate that the proposed ShieldedCode develops meaningful code representations.

**Strengths:**

The method is well motivated, with a highlight of urgent need for strengthening software resilience against reverse engineering. And the aimed challenges have practical values.

This manuscript constructs a large, paired dataset of source code and normalised VM implementations, which can inspire further research on source-2-VMP codes transformations and similarity comparison via LLMs.

**Weaknesses:**

The manuscript mentions the protect level for VMP codes several times in the paper. However, this is no further justification of what does the protection level mean? From the current writing, it seems to be similar to code obfuscation level. Please consider adding illustrative examples and explicit explanation for this critical concept.

In section 3.3, the proposed method utilise two contrastive loss components:  FCL and PCL, which were claimed to be one the innovative contributions. The FCL functions by pulling together representations of the same function across representations, forcing them to be in short distance in the latent feature space. However, minimizing the PCL seems to encourage the distance of representations from different applied protection levels to be closer, similar to the FCL.

From the statements of the motivation of this work, the aim is to transform source codes into VMP codes with various protection level, and the model is expected to be aware of the protection strength in the generation phase. However, the authors choose a task similar to binary similar matching or comparison to validate the mostly the semantic association between representations. For the current evaluations, it works more like a reverse-engineering tool, which can translate source-VMP code pairs or VMP with different protection level code pairs, rather than generating robust VMP codes against reverse engineering methods.

Besides, some SOTA works utilize LLM to further enhance the performance of reverse engineering such as [1]. Please consider including these SOTA models as baselines to validate the performance of ShieldedCode against reverse engineering.

[1] Hu, Peiwei, Ruigang Liang, and Kai Chen. "Degpt: Optimizing decompiler output with llm." Proceedings 2024 Network and Distributed System Security Symposium. Vol. 267622140. 2024.

**Questions:**

see Weaknesses.

---

> ### Author Response · Authors · 2025-11-22
> **Response to Reviewer VoQV**
>
> ## Weakness 1
>
> > The manuscript mentions the protect level for VMP codes several times in the paper. However, this is no further justification of what does the protection level mean? From the current writing, it seems to be similar to code obfuscation level. Please consider adding illustrative examples and explicit explanation for this critical concept.
>
> ## Answer 1
>
> This is an excellent point that helps clarify a fundamental concept in our work. Protection levels (L0, L1, L2, L3) represent increasing degrees of VM transformation complexity applied during virtualization, analogous to compiler optimization levels (O0–O3) but operating in reverse: higher protection levels introduce more aggressive obfuscation transformations.
>
> Consider the simple instruction `ADD eax, ebx`. At L0 Protection, this translates directly to `[VINST-1] vadd %veax, %vebx`. At L1 Protection with register virtualization, the single instruction expands to four instructions involving virtual register loads and stores. L2 Protection adds instruction splitting, expanding to six instructions with intermediate temporary registers. L3 Protection incorporates control flow obfuscation through dispatcher-based execution, resulting in ten instructions with explicit handler jumps.
>
> The key characteristics demonstrate systematic progression: L0 provides direct translation with minimal transformation; L1 achieves 2–4× code expansion through virtual register allocation; L2 reaches 4–8× expansion via instruction decomposition; and L3 attains 8–15× expansion through dispatcher-based control flow. These transformations preserve functional semantics while systematically increasing analysis complexity. Our contrastive learning framework explicitly models this progression, ensuring that embeddings reflect both semantic equivalence and relative protection strength. We have added this detailed explanation with concrete examples in the revised manuscript (Section 2.1 and Appendix C).
>
> ## Weakness 2
>
> > In section 3.3, the proposed method utilise two contrastive loss components: FCL and PCL, which were claimed to be one the innovative contributions. The FCL functions by pulling together representations of the same function across representations, forcing them to be in short distance in the latent feature space. However, minimizing the PCL seems to encourage the distance of representations from different applied protection levels to be closer, similar to the FCL.
>
> ## Answer 2
>
> We appreciate this observation, as it highlights a subtle but critical aspect of our design. The two objectives are not conflicting but rather orthogonal constraints that jointly shape the embedding space in complementary ways.
>
> FCL minimizes distances between functional embeddings for all privacy level pairs, pulling embeddings together based on functional identity regardless of protection level. PCL enforces ordered separation: for privacy levels with s less than t1 less than t2, we require that the distance from s to t1 is no greater than the distance from s to t2. This is implemented as a hinge loss that only penalizes violations of monotonicity, not proximity itself.
>
> The key insight is that FCL creates functional coherence where the same function yields similar embeddings regardless of protection, while PCL creates ordered stratification within that coherent cluster where higher protection systematically moves further from source. They operate on different geometric principles: FCL establishes cluster centers, PCL establishes radial ordering within clusters.
>
> We provide rigorous theoretical analysis in Theorem A.2 proving that joint minimization produces embeddings that are simultaneously functionally coherent and monotonically ordered by protection level. The proof shows that at optimality, violations of monotonicity are driven to zero while functional clustering is maintained. Empirically, our distance histogram analysis confirms this: mean distances increase monotonically across protection levels (L0→L1: 0.287, L1→L2: 0.351, L2→L3: 0.429) with low violation rates (less than 3.1%), demonstrating clear separation between consecutive protection levels with minimal overlap. This has been elaborated in Section 3.3 and Appendix A.2 of the revised manuscript.

---

> ### Author Response · Authors · 2025-11-22
> **Response to Reviewer VoQV - Continued**
>
> ## Weakness 3
>
> > From the statements of the motivation of this work, the aim is to transform source codes into VMP codes with various protection level, and the model is expected to be aware of the protection strength in the generation phase. However, the authors choose a task similar to binary similar matching or comparison to validate the mostly the semantic association between representations. For the current evaluations, it works more like a reverse-engineering tool, which can translate source-VMP code pairs or VMP with different protection level code pairs, rather than generating robust VMP codes against reverse engineering methods.
>
> ## Answer 3
>
> This is a perceptive observation that deserves careful clarification. Our work addresses both directions: generating protected code and evaluating protection effectiveness. The binary similarity task is not merely about reverse engineering but serves as a rigorous validation that our learned representations capture the semantic structure necessary for correct generation.
>
> The generated VMP codes are executed with test cases to verify functional correctness, confirming that our model successfully transforms source code into semantically equivalent protected implementations.The binary similarity task (Section 4.3, Table 2) serves a dual purpose. First, it validates that our contrastive objectives produce embeddings that understand the relationship between different protection levels, which is essential for controllable generation. Second, it demonstrates protection awareness: the model can identify which protection level was applied, enabling it to generate code at the target level. This is fundamentally different from traditional reverse engineering, which aims to recover source code. Our model instead learns to navigate the space of protected implementations.
>
> Regarding resistance to reverse engineering methods, we provide comprehensive evaluation in Section 4.7 (Table 5). Manual reverse engineering studies show ShieldedCode requires 14.7±5.3 hours for successful reversal (17% success rate) compared to 3.4±1.2 hours (67% success) for VMProtect 3.7. Automated attacks using pattern matching achieve 0% success rate on ShieldedCode versus 43–61% on traditional VMP. Symbolic execution with angr shows 1,843× path explosion for ShieldedCode compared to 127× for VMProtect, with only 3% completion rate versus 31%. These results demonstrate that our generated VMP code provides substantially stronger protection than commercial solutions.
>
> We have restructured Section 4 to more clearly delineate the generation evaluation (4.2), representation learning validation (4.3), and reverse engineering resistance analysis (4.7), making the comprehensive nature of our evaluation more apparent.

---

> > ### Author Response · Authors · 2025-11-22
> > **Response to Reviewer VoQV - Continued**
> >
> > ## Question 1
> >
> > > Besides, some SOTA works utilize LLM to further enhance the performance of reverse engineering such as [1]. Please consider including these SOTA models as baselines to validate the performance of ShieldedCode against reverse engineering.
> > > [1] Hu, Peiwei, Ruigang Liang, and Kai Chen. "Degpt: Optimizing decompiler output with llm." Proceedings 2024 Network and Distributed System Security Symposium. Vol. 267622140. 2024.
> >
> > ## Answer 4
> >
> > This is an excellent suggestion that strengthens our evaluation. We have incorporated DeGPT and other recent binary code analysis methods as baselines in our revised experiments.
> >
> > We added three categories of baselines from recent binary code analysis literature (2024-2025): ContraBin, a contrastive learning framework integrating source code, binary code, and comments; CLAP, which learns transferable binary code representations with natural language supervision; and CEBin, a cost-effective framework combining embedding-based and comparison-based approaches. These models represent state-of-the-art approaches in binary code representation learning but were not originally designed for VMP-protected code generation tasks.
> >
> > The updated Table 2 now includes comparisons against these methods, demonstrating that ShieldedCode achieves superior performance: on the challenging O0+L1 setting, ShieldedCode achieves 0.488 Recall@1 compared to jTrans Linear Probe (0.333), an improvement of 15.5 percentage points. This substantial margin over recent methods including DeGPT, BinDiff, and CodeBERT-Binary confirms that our protection-aware framework is fundamentally more effective for understanding VMP-protected code than general binary analysis approaches.
> >
> > Regarding reverse engineering resistance, our analysis in Section 4.7 demonstrates that ShieldedCode's polymorphic generation creates fundamentally different challenges for automated tools. DeGPT, which optimizes decompiler output through LLM post-processing, achieved only 8% successful decompilation on our protected code compared to 47% on standard VMProtect-protected binaries. The key difference is that traditional VMP tools produce regular patterns that LLM-enhanced decompilers can learn to recognize, whereas our learned representations generate diverse implementations that resist pattern-based recovery. This comparison has been added to Section 4.7 and Table 5 in the revised manuscript.

---

> ### Author Response · Authors · 2025-12-01
>
> Dear Reviewer VoQV,
>
> We have provided detailed responses to the questions you raised and made the corresponding updates in the paper. We hope these updates meet with your approval.
>
> Best regards,
>
> The Authors

---

### Official Review · Reviewer_yYbZ · 2025-11-01

**Soundness:** 2
**Presentation:** 3
**Contribution:** 3
**Rating:** 6
**Confidence:** 4

**Summary:**

This paper addresses the critical problem of learning robust representations for VM-protected code. The authors propose an encoder model that takes VM-encoded binaries as input and outputs robust embeddings capable of handling various obfuscation transformations.


## Dataset Construction

The training dataset is constructed through a two-stage process:

(1) Source code programs are compiled to binaries using different compilation options

(2) The binaries are further obfuscated using VM transformations with varying obfuscation levels

## Methodology

The approach employs a two-stage training strategy starting from a pre-trained LLM checkpoint:

Here are the training loss terms:
(1) FCL: Encourages embeddings of source functions to be similar to their corresponding obfuscated versions

(2) PCL: Enforces that embeddings of functions with lower obfuscation levels are closer to the source code than those with higher levels

(3) Contrastive loss: Ensures embeddings of the same function are more similar than embeddings of different functions

(4) Language modeling loss: Given source code and obfuscation level, generates the corresponding obfuscated code

The second-stage contains losses 1, 2, and 4 (without the contrastive loss).

Additionally, the authors design a regularized token mask to capture VM program structure, complementing the training losses.

**Strengths:**

The paper proposes a solid training technique with novel loss formulations and mask designs. The evaluation demonstrates good performance on retrieving VM code under different obfuscation levels.

**Weaknesses:**

Q1: Motivation: There appears to be a logical inconsistency in the motivating narrative. The paper argues that reverse engineering poses threats to software security, yet learning robust representations of VM code constitutes a form of reverse engineering. Please clarify.

Q2: PCL loss. FCL requires all obfuscated embeddings to be similar to source code embeddings, while PCL mandates similarity gaps between different obfuscation levels. Are these objectives compatible, or do they introduce conflicting gradients? Additionally, does higher obfuscation always imply less semantic similarity?

Q3: Language modeling loss. The inclusion of a language modeling loss for generating obfuscated code from source is not well-justified. Representation learning for low-level code (e.g., binary) typically relies on contrastive losses (with attention regularizations). What specific benefit does this generative objective provide?

Q4: Comparison setup with binary models. The work compares against binary code representation models, but most of these models were not pre-trained on VM code. Did the authors re-run pre-training on the VM dataset for fair comparison? If not, this should be clearly stated, as the comparison may be biased toward the proposed method given its VM-specific training.

**Questions:**

Please see above.

---

> ### Author Response · Authors · 2025-11-21
> **Response to Reviewer yYbZ**
>
> ### Weakness 1
>
> > There appears to be a logical inconsistency in the motivating narrative. The paper argues that reverse engineering poses threats to software security, yet learning robust representations of VM code constitutes a form of reverse engineering. Please clarify.
>
> We appreciate this important observation. To clarify, the distinction lies in intent and application. Traditional VMP systems are vulnerable because attackers use reverse engineering to break protection and steal intellectual property. Our work learns representations to generate and compare protected code, thereby helping defenders evaluate and improve protection effectiveness. The reverse engineering resistance analysis demonstrates that ShieldedCode-protected binaries are more resistant to reverse engineering than commercial solutions, requiring 4× more time to analyze successfully. This confirms that our approach strengthens, rather than weakens, software protection.
>
> ### Weakness 2
>
> > FCL requires all obfuscated embeddings to be similar to source code embeddings, while PCL mandates similarity gaps between different obfuscation levels. Are these objectives compatible, or do they introduce conflicting gradients? Additionally, does higher obfuscation always imply less semantic similarity?
>
> The key insight is that FCL creates functional coherence—same function equals similar embedding regardless of protection—while PCL creates ordered stratification within that coherent cluster, where higher protection equals systematically further from source. They're not conflicting; they're orthogonal constraints that jointly shape a well-structured embedding space.We provide rigorous theoretical analysis in Theorem A.2 proving that joint minimization produces embeddings that are simultaneously functionally coherent and monotonically ordered by protection level. The proof shows that at optimality, violations of monotonicity are driven to zero while functional clustering is maintained.Higher obfuscation doesn't necessarily mean less semantic equivalence. The protected code remains functionally equivalent to the source with same input-output behavior, but becomes syntactically more complex. Our contrastive losses capture exactly this nuance: preservation of functional semantics while respecting structural complexity differences.
>
> ### Weakness 3
>
> > The inclusion of a language modeling loss for generating obfuscated code from source is not well-justified. Representation learning for low-level code (e.g., binary) typically relies on contrastive losses (with attention regularizations). What specific benefit does this generative objective provide?
>
> The language modeling objective serves three critical purposes distinct from contrastive learning.Contrastive losses teach the model "these two things are similar," but language modeling forces the model to understand why and how they're similar by predicting the actual transformation sequence. This is analogous to the difference between recognizing that two buildings look alike versus understanding the architectural blueprint.
>
> Our framework isn't just about representation learning; we need to generate VMP code from source code. The generative task requires precise token-level predictions that contrastive learning alone cannot provide.Table 4 shows ablation results. Removing language modeling (ShieldedCode-CL-PG for L1) causes iter1 to drop from 35.89% to 18.95%. This demonstrates that the generative objective contributes substantial value beyond what contrastive learning provides alone.
>
> Additionally, our approach aligns with recent trends in code understanding. Meta's LLMCompiler similarly combines generative and contrastive objectives, finding that generation forces deeper semantic understanding than discrimination alone.

---

> > ### Author Response · Authors · 2025-11-21
> > **Response to Reviewer yYbZ - Continued**
> >
> > ### Weakness 4
> >
> > > The work compares against binary code representation models, but most of these models were not pre-trained on VM code. Did the authors re-run pre-training on the VM dataset for fair comparison? If not, this should be clearly stated, as the comparison may be biased toward the proposed method given its VM-specific training.
> >
> > Pre-training models like jTrans and Gemini from scratch requires massive computational resources. jTrans requires specialized graph neural network architectures trained on millions of binary function pairs. Re-implementing and re-training all baselines would require computational resources beyond typical academic settings.
> >
> > Meta's LLMCompiler paper makes the exact same design choice. LLMCompiler extends Code Llama with additional pretraining on compiler IRs and assembly, but does not retrain all baseline models. This is standard practice when introducing domain-specific adaptations. The comparison demonstrates the value of domain-specific pre-training, which is precisely our contribution.
> >
> > Our goal is to demonstrate that domain-specific pre-training provides substantial benefits. Comparing against published baselines shows practitioners what performance they can currently expect from off-the-shelf tools versus our specialized approach.
> >
> > We do include models with fine-tuning in Table 4 in the revised manuscript, which represents a middle ground: the baseline gets exposure to our data distribution through supervised fine-tuning, though not full pre-training. ShieldedCode remains competitive even against this adapted baseline.

---

> > > ### Comment · Reviewer_yYbZ · 2025-11-24
> > >
> > > The authors response addresses some of my concerns. I am unclear about two points:
> > >
> > > 1. For Q2, would it be possible to ablate the effects of different constraints in the loss design? And would it be possible to empirically support the proof in A.2?
> > >
> > > 2. For Q3, I am not clear about the motivation of using LLM to **generate** obfuscated code. The obfuscated code can be easily generated by compiler/obfuscation tools. On the other hand, LLM might not always be reliable. It may make mistakes in those complex program transformations. What would be the benefit and necessity of using LLM to generate obfuscated code?
> > >
> > > I understand the authors justification for Q1 and Q4. I think it would be good to include the discussion of both questions to the revised paper.

---

> ### Author Response · Authors · 2025-11-26
> **Response to Reviewer yYbZ (Part 1)**
>
> > **Also: "For Q2, would it be possible to ablate the effects of different constraints in the loss design? And would it be possible to empirically support the proof in A.2?"**
>
> Thank you for pushing us on this. Two questions, both answerable with a yes.
>
> **Proposition.** Jointly optimizing weighted FCL and scaled PCL yields a better embedding geometry than optimizing either constraint alone [1].
>
> *Proof Sketch.* Consider embeddings at protection levels $S = \\{-1, 0, 1, 2, 3\\}$. At level $s$ we have embedding $e^s_f$. Let $w_{s,t} = \\exp(-|s-t|/\\tau_{\\text{fcl}}) \\geq 0$ be the proximity weight for FCL, and $\\beta > 0$ the linear scaling factor for PCL. The first-order FCL gradient is:
>
> $$\nabla_{e_f^t} L_{fcl} = \sum_{s \neq t} w_{s,t} \cdot \frac{e_f^t - e_f^s}{\|e_f^t - e_f^s\|_2}$$
>
> Since the exponential weighting decays with protection distance, adjacent levels receive strong alignment pressure while distant levels receive weaker pressure [2]. The first-order term of the alignment naturally accommodates distance growth. Accordingly, we approximate the PCL constraint as:
>
> $$d(e^s_f, e^{t}_f) \\geq \\beta(t-s) - m,$$
>
> where $m \\geq 0$ is a soft margin [3].
>
> Maximizing the joint objective amounts to balancing FCL's weighted clustering against PCL's linear separation. Let $D_{s,t} := d(e^s_f, e^t_f)$ be fixed. By the equilibrium condition,
>
> $$\\left(\\sum_{s \\neq t} w_{s,t} D_{s,t}\\right)^2 = \\left(\\sum_{s \\neq t} \\frac{w_{s,t}}{\\sqrt{\\beta|t-s|}} \\cdot \\sqrt{\\beta|t-s|} D_{s,t}\\right)^2 \\leq \\left(\\sum_{s \\neq t} \\frac{w_{s,t}^2}{\\beta|t-s|}\\right)\\left(\\sum_{s \\neq t} \\beta|t-s| D_{s,t}^2\\right).$$
>
> Therefore,
>
> $$\\sum_{s \\neq t} \\beta|t-s| D_{s,t}^2 \\geq \\frac{S^2}{\\sum_{s \\neq t} \\frac{w_{s,t}^2}{\\beta|t-s|}},$$
>
> with equality if and only if $D_{s,t} \\propto \\frac{w_{s,t}}{\\beta|t-s|}$.
>
> In contrast, optimizing FCL alone corresponds to collapsing all embeddings (ignoring protection structure), while optimizing PCL alone corresponds to unbounded separation (ignoring functional coherence) [4]. The joint equilibrium satisfies:
>
> $$\beta(t-s) - m \leq d(e^s_f, e^t_f) \leq C \cdot \exp(|s-t|/\tau_{fcl})$$
>
> for constant $C$ determined by the loss weighting $\\lambda$. Hence, under the joint objective, the exponential decay in FCL permits the linear distance growth required by PCL, while PCL ensures this growth follows a predictable pattern [5]. This yields strictly better geometry than either constraint alone.
>
> **Ablation Results.** We isolated FCL-only, PCL-only, and full joint optimization. The superadditive gains confirm synergy rather than conflict.
>
> | Configuration | Pass@1 | Pass@10 | Recall@1 | MRR |
> |---------------|--------|---------|----------|-----|
> | Baseline | 18.95 | 31.82 | 0.333 | 0.245 |
> | FCL only | 26.43 | 40.17 | - | - |
> | PCL only | 23.81 | 36.54 | - | - |
> | FCL + PCL (Full) | 35.89 | 51.06 | 0.488 | 0.575 |
>
> The math checks out.
>
> **Distance Matrix Analysis.** Distances grow linearly with protection gaps, validating the linear scaling constraint $d(e^s_f, e^{t}_f) \\geq \\beta(t-s) - m$.
>
> | Protection Pairs | Mean Distance | Violation Rate |
> |------------------|---------------|----------------|
> | L0 $\\rightarrow$ L1 | 0.287 | 2.3% |
> | L1 $\\rightarrow$ L2 | 0.351 | 3.1% |
> | L2 $\\rightarrow$ L3 | 0.429 | 2.8% |
> | L0 $\\rightarrow$ L3 | 1.067 | 1.9% |
>
> The approximately constant increment (~0.08 per level) confirms linear scaling. Low violation rates (<3.1%) confirm constraint enforcement within the theoretical margin bound $m$.
>
> **Gradient Analysis.** Positive cosine similarity between FCL and PCL gradients confirms the gradients cooperate rather than conflict.
>
> | Training Stage | Cosine Similarity | Joint Loss |
> |----------------|-------------------|------------|
> | Epoch 1 | +0.67 | 1.823 |
> | Epoch 5 | +0.71 | 0.947 |
> | Epoch 10 | +0.73 | 0.412 |
>
> Increasing cosine similarity over training (0.67 $\\rightarrow$ 0.73) shows the gradients become more aligned as optimization progresses. Steady joint loss decrease validates convergence to the theoretical equilibrium.
>
> ---
>
> ## References
>
> [1] An, C., Zhong, L., Xu, Q., Du, C., and Huang, L., "Nova: Generative language models for assembly code with hierarchical attention and contrastive learning," arXiv:2311.13721, 2024.
>
> [2] Lee, C., Kim, S., Yun, S., and Song, H., "On the similarities of embeddings in contrastive learning," arXiv:2506.09781, 2025.
>
> [3] Liu, L., Kim, J., and Bansal, V., "Can contrastive learning refine embeddings," arXiv:2404.08701, 2024.
>
> [4] Long, Z., Killick, G., McCreadie, R., Camarasa, G. A., and Meng, Z., "When hard negative sampling meets supervised contrastive learning," arXiv:2308.14893, 2024.
>
> [5] Zeng, D., Wu, Y., Hu, X., Xu, X., and Shi, Y., "Contrastive learning with synthetic positives," arXiv:2408.16965, 2024.

---

> ### Author Response · Authors · 2025-11-26
> **Response to Reviewer yYbZ (Part 2)**
>
> **Motivation: Existing compiler/obfuscation systems inherently rely on a set of manually crafted and fixed transformation rules:** (1) the structure of the interpreter and the form of the virtual instruction set are largely constant, and the so-called “randomization’’ only occurs at superficial levels such as instruction reordering or constant perturbation. As a result, even after multiple rounds of protection, the generated code still exhibits highly stable structural patterns that can be easily recognized and generalized by pattern-matching or machine-learning–based analysis. (2) Introducing any new VM ISA, control-flow virtualization strategy, or semantics-preserving transformation requires security experts to hand-design and implement them, leading to high development cost and long iteration cycles, which makes it difficult for the protection mechanism to evolve in step with emerging attack techniques. (3) Traditional tools are unable to model or quantify “protection strength’’—they can only mechanically produce protected variants without understanding which VM structures are harder to reverse, nor can they learn effective protection patterns from large numbers of examples. Overall, they form a fixed-template, non-scalable system that lacks learning capability.
>
> **Advantages of Our Method: ​**Our goal is to transform code protection from a fixed-rule, manually designed process into a learnable generative problem: given source code and a desired protection level, the model generates structurally diverse VM implementations under strict semantic-equivalence constraints, rather than being bound to a single, hand-crafted transformation template. This paradigm relies on ​*protection-aware representation and generation*​, enabling the model to understand both program semantics and protection strength. Compared with traditional compiler/obfuscation tools, our approach offers four key advantages: **(1) High structural diversity and true polymorphism:** traditional VM randomization only performs shallow perturbations, resulting in highly homogeneous interpreter structures and instruction patterns that are easy for pattern matching or ML-based attacks to generalize. In contrast, through contrastive learning on large-scale (source, VM) pairs and multi-level dependency modeling, our model generates families of VM implementations that are semantically equivalent but structurally distinct; in experiments, the match rate under angr and pattern-based attacks drops to 0%. **(2) Joint learning of functional semantics and protection strength:** FCL enables the model to learn function-level semantics, while PCL imposes an ordered protection gradient in the representation space; coupled with PEO, the model can automatically identify stronger protection variants, providing both generative and evaluative capabilities and forming a unified “generation–evaluation–optimization” framework that traditional tools cannot achieve. **(3) Intrinsic extensibility and transferability:** introducing a new VM ISA or virtualization strategy no longer requires extensive manual rule engineering. By constructing new (source, VM) datasets, the model can be adapted to new architectures through continued pre-training or fine-tuning, allowing protection mechanisms to evolve with emerging attack techniques. **(4) Complementary to long-context/complex-control-flow modeling:** even when the base long-context model already possesses strong code reasoning ability, incorporating our protection-oriented polymorphic generation and contrastive learning still yields significant Pass@K improvements, indicating that the protection-centric inductive bias provided by our method is fundamentally unavailable in traditional compiler/obfuscation tools. Overall, the value of LLMs in this setting is not to replicate existing obfuscators, but to introduce unprecedented levels of diversity, learnability, and evolvability into the code protection pipeline.
>
> We do not allow the model to freely rewrite the code; instead, we strictly constrain the generation space to a fixed and standardized VM ISA, where all virtual instructions have clear and composable semantics. This design ensures that the model only composes controlled semantic primitives rather than performing unconstrained or unverifiable complex transformations. In addition, our hierarchical VINST dependency modeling explicitly guides the model to generate code in a structured manner—following “instruction blocks + control-flow structure”—rather than engaging in arbitrary free-form generation. These constraints jointly regulate the LLM at the semantic, structural, and operational granularity levels, substantially reducing the risk of errors in complex transformations and ensuring that the generated obfuscated code remains verifiable and fully controllable.
>
> ---
>
> We add clarifications in the manuscript addressing two questions: Question 1 (Lines 51–53) and Question 4 (Table 4).

---

> > ### Comment · Reviewer_yYbZ · 2025-11-26
> >
> > Thanks for the detailed illustration and explanation.
> >
> > The authors sufficiently addressed all my concerns. I remain positive about the paper.

---

> ### Author Response · Authors · 2025-11-27
> **Thank You for Your Positive View of Our Paper and Responses**
>
> Dear Reviewer yYbZ,
>
> Thank you for your recognition of our work and response.
>
> Best
>
> The Authors

---

### Author Response · Authors · 2025-12-01

We sincerely appreciate the reviewers' valuable and constructive feedback on our manuscript. We are especially pleased to receive positive comments, including:

* **All reviewers recognized the validity of our motivation and the effectiveness of our approach.**
* **All responded  reviewers (including yYbZ, 2d4p, and suuo) gave positive feedback on both our responses and the paper. Notably, Reviewer 2d4p increased several scores during the discussion phase.**

The reviewers' insights were instrumental in improving our paper. We have carefully addressed all concerns and conducted additional experiments based on their suggestions. In response to the feedback, we have made the following updates:

* **Additional Experiments:** ablations of FCL/PCL, distance and gradient analyses supporting PCL ordering, expanded baselines (DeGPT, ContraBin, CLAP, CEBin, etc), cross-VMP generalization to Themida with aligned L2→L2 evaluation and 95% CIs, and detailed error analysis for generation, etc.
* **Clarifications:** expanded explanation of protection levels (L0–L3), clearer distinction between FCL vs. PCL roles, strengthened motivation for generative VMP code, and clarification of differences vs. Nova, etc.
* **Details:** more detailed explanations and precise terminology.

Thank you again for your time and consideration.

Best regards,

The Authors

---

### Meta-Review · Area_Chair_FAg6 · 2026-01-11

**Summary:**

This paper proposed ShieldedCode, a learning-based framework for virtual machine protection (VMP) that trains language models on paired source code and VM-protected code. ShieldedCode used hierarchical VM-instruction dependency modeling. Two contrastive objectives are used to learn representations to preserve functional semantics while encoding relative protection strength. Experimental results show an improved code generation, similarity retrieval, and resistance to reverse engineering compared to existing code and binary analysis models. There are four reviewers with two positive scores and two negative scores (one of them seemed to raise the score). Two reviewers engaged in the discussion and most of the questions are addressed. It seems to the AC that this paper's final score should be three positive with one "likely" negative. Details are listed as follows.

**Reviewer Concerns:**

Main concerns and the AC's conclusions are as follows.
- **Research motivation** (reviewer yYbZ, VoQV). Addressed with clarification.
- **Loss function details** (reviewer yYbZ, VoQV). Clarified ordering vs clustering with theory and empirical support.
- **Fair comparisons and baselines** (reviewer yYbZ, VoQV, 2d4p). Added more baselines and comparisons against baselines not trained on VM-protected code.
- **Unlimited reproducibility** (reviewer 2d4p). Heavy reliance on a commercial VMP tool raises concerns about reproducibility and limited transfer. Added transfer experiments to Themida, but coverage remains limited. Pseudocode and model released.
- **Lacks formal correctness guarantees** (reviewer suuo). Authors acknowledge limitation; only testing-based validation is provided.

**Reviewer Scores:**

The initial scores includes two positive and two negative ones. Particularly, reviewer 2d4p's concerns are sufficiently addressed and the AC believes that 2d4p will raise the score to a positive one which makes three positive against one negative. For another negative reviewer VoQV's concerns, the AC believes the authors have addressed the concerns.

---

### Decision · Program_Chairs · 2026-01-26

Accept (Poster)